# Ocean warming affected faunal dynamics of benthic invertebrate assemblages across the Toarcian Oceanic Anoxic Event in the Iberian Basin (Spain)

**Veronica Piazza**[1]*, **Clemens V. Ullmann**[2], **Martin Aberhan**[1]

**1** Museum für Naturkunde, Leibniz Institute for Evolution and Biodiversity Science, Berlin, Germany,
**2** University of Exeter, Camborne School of Mines, College of Engineering, Mathematics and Physical Sciences, Penryn, Cornwall, United Kingdom

* Veronica.Piazza@mfn.berlin, freyja29@gmail.com

**Data Availability Statement:** The datasets generated and analysed during the current study, as well as the R scripts, are available from the

## Abstract

The Toarcian Oceanic Anoxic Event (TOAE; Early Jurassic, ca. 182 Ma ago) represents one of the major environmental disturbances of the Mesozoic and is associated with global warming, widespread anoxia, and a severe perturbation of the global carbon cycle. Warming-related dysoxia-anoxia has long been considered the main cause of elevated marine extinction rates, although extinctions have been recorded also in environments without evidence for deoxygenation. We addressed the role of warming and disturbance of the carbon cycle in an oxygenated habitat in the Iberian Basin, Spain, by correlating high resolution quantitative faunal occurrences of early Toarcian benthic marine invertebrates with geochemical proxy data ($\delta^{18}O$ and $\delta^{13}C$). We find that temperature, as derived from the $\delta^{18}O$ record of shells, is significantly correlated with taxonomic and functional diversity and ecological composition, whereas we find no evidence to link carbon cycle variations to the faunal patterns. The local faunal assemblages before and after the TOAE are taxonomically and ecologically distinct. Most ecological change occurred at the onset of the TOAE, synchronous with an increase in water temperatures, and involved declines in multiple diversity metrics, abundance, and biomass. The TOAE interval experienced a complete turnover of brachiopods and a predominance of opportunistic species, which underscores the generality of this pattern recorded elsewhere in the western Tethys Ocean. Ecological instability during the TOAE is indicated by distinct fluctuations in diversity and in the relative abundance of individual modes of life. Local recovery to ecologically stable and diverse post-TOAE faunal assemblages occurred rapidly at the end of the TOAE, synchronous with decreasing water temperatures. Because oxygen-depleted conditions prevailed in many other regions during the TOAE, this study demonstrates that multiple mechanisms can be operating simultaneously with different relative contributions in different parts of the ocean.

Dryad Digital Repository (accession number: doi:10.5061/dryad.66t1g1k0w).

**Funding:** V.P. and M.A. were funded by the Deutsche Forschungsgemeinschaft grant DFG AB 09/10-1. C.V.U acknowledges funding from the JET project (NERC grant NE/N018508/1). The funders had no role in study design, data collection and analysis, decision to publish, or preparation of the manuscript.

**Competing interests:** The authors have declared that no competing interests exist.

# Introduction

The early Toarcian Oceanic Anoxic Event (TOAE, ca. 182 Ma) [1], is one of the major carbon cycle perturbations of the Mesozoic [2], marked globally in the sedimentary, geochemical and paleontological records (e.g., [3,4]). Its ultimate cause is most likely the rapid and extensive increase in $pCO_2$ triggered by large-scale eruptions of the Karoo-Ferrar igneous province [5] and methane release from both oceanic [6] and terrestrial sources [7]. Consequences for marine ecosystems included increased water temperatures, widespread anoxia evidenced by black shale deposition, and possibly ocean acidification. The TOAE thus provides an opportunity to investigate the effects of climate warming on marine communities as recorded by fossils and may serve as a potential deep-time analog for the consequences of current and projected climate change on extant communities. The magnitude of seawater warming is estimated to have ranged between +3˚C and +7˚C, depending on latitude [8–12]. While dysoxic–anoxic conditions and black shales are typical for the TOAE in many regions, they can be very limited in extent and duration or even absent in other areas such as SW Europe and northern Africa [9,11,13–18]. This implies different regional expressions of the TOAE [19,20]. Additional inferred effects of elevated $pCO_2$ are increased continental weathering rates [21], ocean acidification [22,23] and a crisis in carbonate production [24].

The patterns and rates of ecosystem recovery from environmental stress during mass extinctions are as yet insufficiently studied [25]. However, understanding community stability when facing disturbances is central to inform present-day conservation management [26]. Biodiversity loss and extinctions in fossil communities might provide a long-term perspective of ecosystem response to global warming. The TOAE coincides with elevated extinction rates for nektonic and benthic organisms, i.e. molluscs, brachiopods, ostracods, foraminifera and dinoflagellates, and with changes in community structure (e.g., [18,24,27–33]. The proximate causes are debated and most studies see widespread dysoxia–anoxia as the main stressor (e.g., [5,24,27–30,34–37], with Them et al. [38] even arguing for global ocean deoxygenation to have already started at the Pliensbachian/Toarcian boundary. Yet, the record of dysoxia–anoxia is far from ubiquitous as is evident from faunal and facies analyses in oxygenated environments from SW Europe [9,11,13–15,17,39]. Even from restricted basins where anoxia developed, oxygenated bottom conditions are recorded before the onset of the TOAE (e.g., [29]). In the absence of low-oxygen conditions, warming was suggested as an important stressor (e.g., [11,40–43]), but rigorous statistical tests of the relationship between temperature change and faunal change are few (e.g., [9,44–46]).

Experiments have shown that raised temperatures can lead to physiological stress for the marine ectotherms that are above their thermal optimum, through increased metabolic oxygen demand and reduced oxygen transport (e.g., [47,48]). Thus, temperature-driven stress may decrease population performance, leading to extirpations and ultimately to extinction [49]. What seems clear from the debate on the TOAE is that there is no universal trigger of the biotic crisis, and that regional differences in environmental conditions are important. As dysoxia–anoxia is not a plausible main stressor in oxygenated settings, the role of other factors should be tested statistically.

Here, we investigated the taxonomic and ecological responses of marine benthic macroinvertebrate assemblages across the TOAE in an oxygenated setting of the Iberian Basin, Spain, with three objectives: (i) Establish the patterns, timing, and magnitude of change in taxonomic composition and ecological structure of fossil communities from pre-TOAE times into the TOAE and post-TOAE intervals; (ii) test for negative faunal effects of elevated ocean temperatures and perturbations of the carbon cycle as recorded by geochemical proxy data; and (iii)

interpret the results in terms of ecosystem stability, faunal recovery, and the relationship between temperature and faunal change.

## Geological framework

Faunal and geochemical data were collected from the same samples along the sedimentary succession of Barranco de la Cañada (40˚23'53.4"N 1˚30'07.4"W), located near Albarracín, in the Iberian Basin, Spain (Fig 1). This section was chosen for its well-preserved and abundant benthic macroinvertebrate fauna, its relative stratigraphic completeness and its biostratigraphic control based on brachiopods and ammonites [14,50].

Ca. 28 m of the succession were sampled, spanning from the Pliensbachian/Toarcian boundary to the lower Bifrons Zone (middle Toarcian) (see sedimentary log in Fig 2). This time span corresponds to the Turmiel Formation [52], which is characterized by alternations of marlstones and argillaceous limestones. The depositional environment is interpreted as a low-energy, mid-ramp setting at 40–70 m depth, below storm wave base [14,15,53]. Rare packstones and rudstones indicate episodes of increased water energy. The absence of black shales, the relatively abundant fauna with widespread bioturbation across the TOAE interval [20,54] and the generally low TOC (<1.2 wt. %) measured for the formation [40] are evidence of oxygenated bottom conditions during the event. The sequence stratigraphic architecture of the

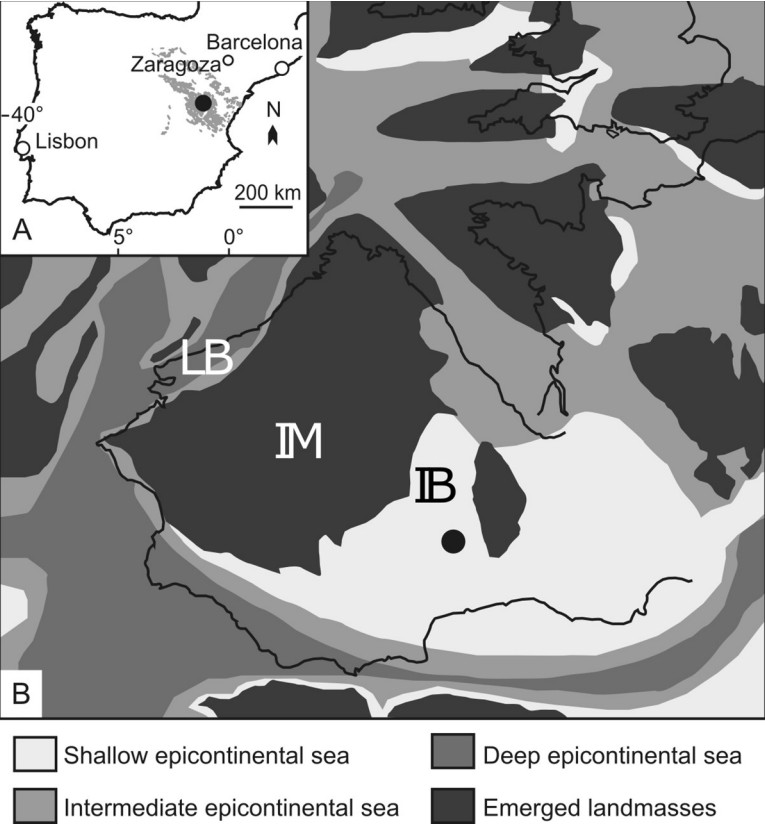

**Fig 1. Geographical location and paleogeographical position of Barranco de la Cañada.** Location map of the investigated section in the Iberian Range system (A) and paleogeographical reconstruction of the western Tethyan basins (B). Studied area marked by the black dot. The grey shading in (A) represents the Iberian Chain. Abbreviations in (B): LB = Lusitanian Basin; IM = Iberian Massif; IB = Iberian Basin. (A) and (B) modified respectively after Figs 1 and 6 in ref. [51] under a CC BY license, with permission from Elsevier, original copyright 2015.

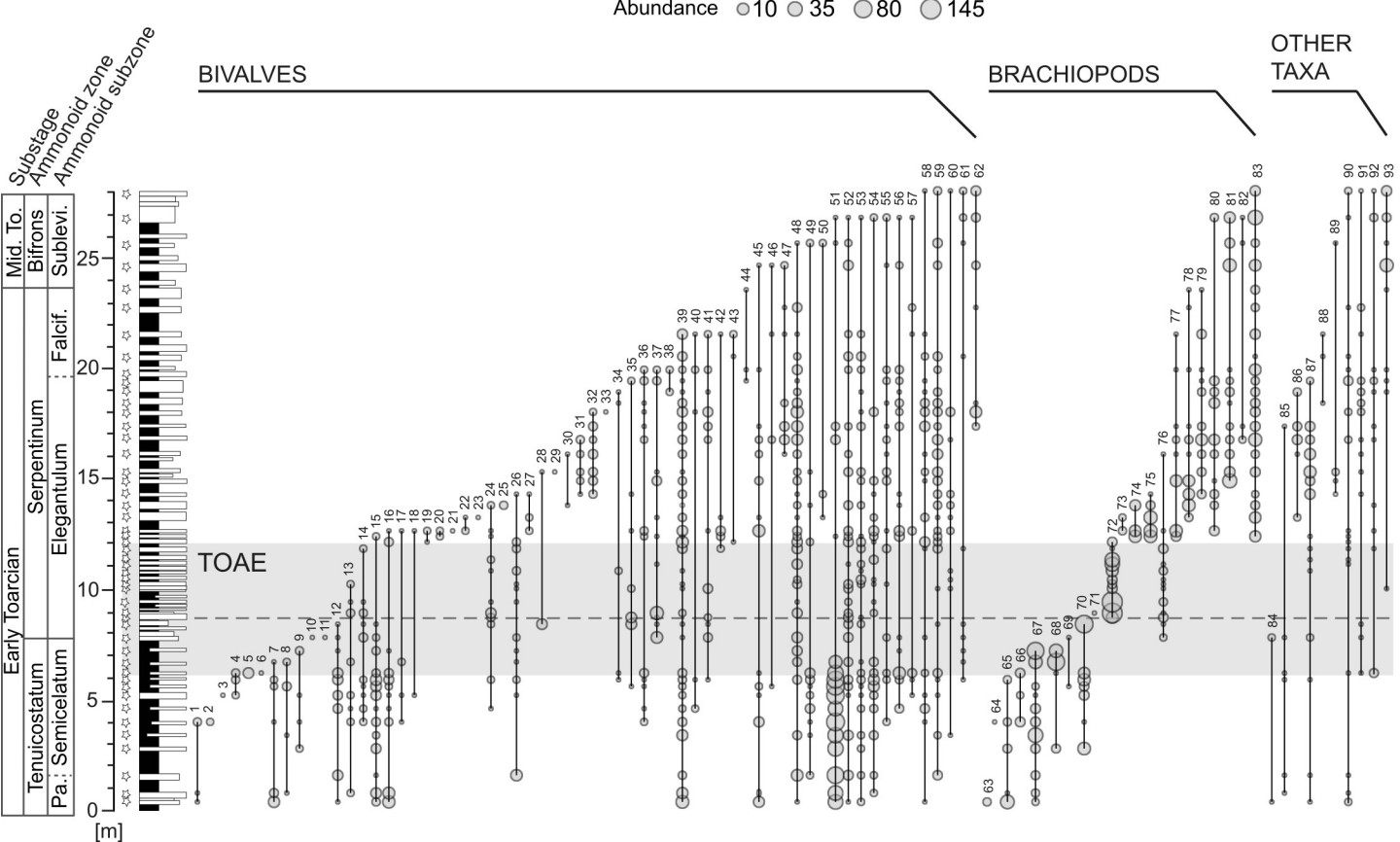

**Fig 2. Stratigraphy and ranges of benthic macroinvertebrate species at Barranco de la Cañada.** Taxa are ordered by last occurrences, and subdivided into taxonomical groups. Each circle represents the presence of a taxon at a sampled level, the size reflecting the abundance of the taxon. The star-shaped symbols indicate the sampled levels. The shaded grey area represents the extent of the chemostratigraphically defined Toarcian Oceanic Anoxic Event (TOAE) interval. Chronostratigraphic boundaries modified from ref. [14]. The horizontal dashed line marks the level with the last appearances of the pre-TOAE brachiopod fauna.

basin has been subdivided into second and third order cycles [14,15,53,55]. The studied interval can be assigned to the second-order transgressive-regressive cycle termed LJ-3 in ref. [53] (Fig 3), which spans from the early Pliensbachian to the middle Toarcian.

The succession at Barranco de la Cañada shows the well-known TOAE negative carbon isotopic excursion which ranges from the uppermost Tenuicostatum Zone up to the lower Serpentinum Zone ([57], Fig 3). This negative carbon isotope excursion is recorded globally from bulk rock carbonates, organic matter, wood and biogenic calcite (e.g., [12,22,59,60]). Its duration is estimated as ∼300 to 500 kyr according to recent studies [61,62], but uncertainty around those numbers remains (see ref. [63] for a summary). These changes in carbon isotope ratios reflect changes in the global carbon cycle, but the causes of the variation are complex, including the release of isotopically light volcanic, thermogenic, and/or biogenic carbon into the global ocean–atmosphere system, changes in primary productivity, and changes in the rates of organic matter burial [4].

Oxygen isotope ratios are available from the same samples at Barranco de la Cañada ([57]; see 'Material and methods' below). δ18O values in shells are controlled by ambient water temperatures and other factors [58] but, in the studied material, they reliably represent temperature signals ([57]; see 'Discussion'). Accordingly, early Toarcian temperature increased rapidly at the onset of the TOAE and stayed high throughout the interval (Fig 3). Local bottom water

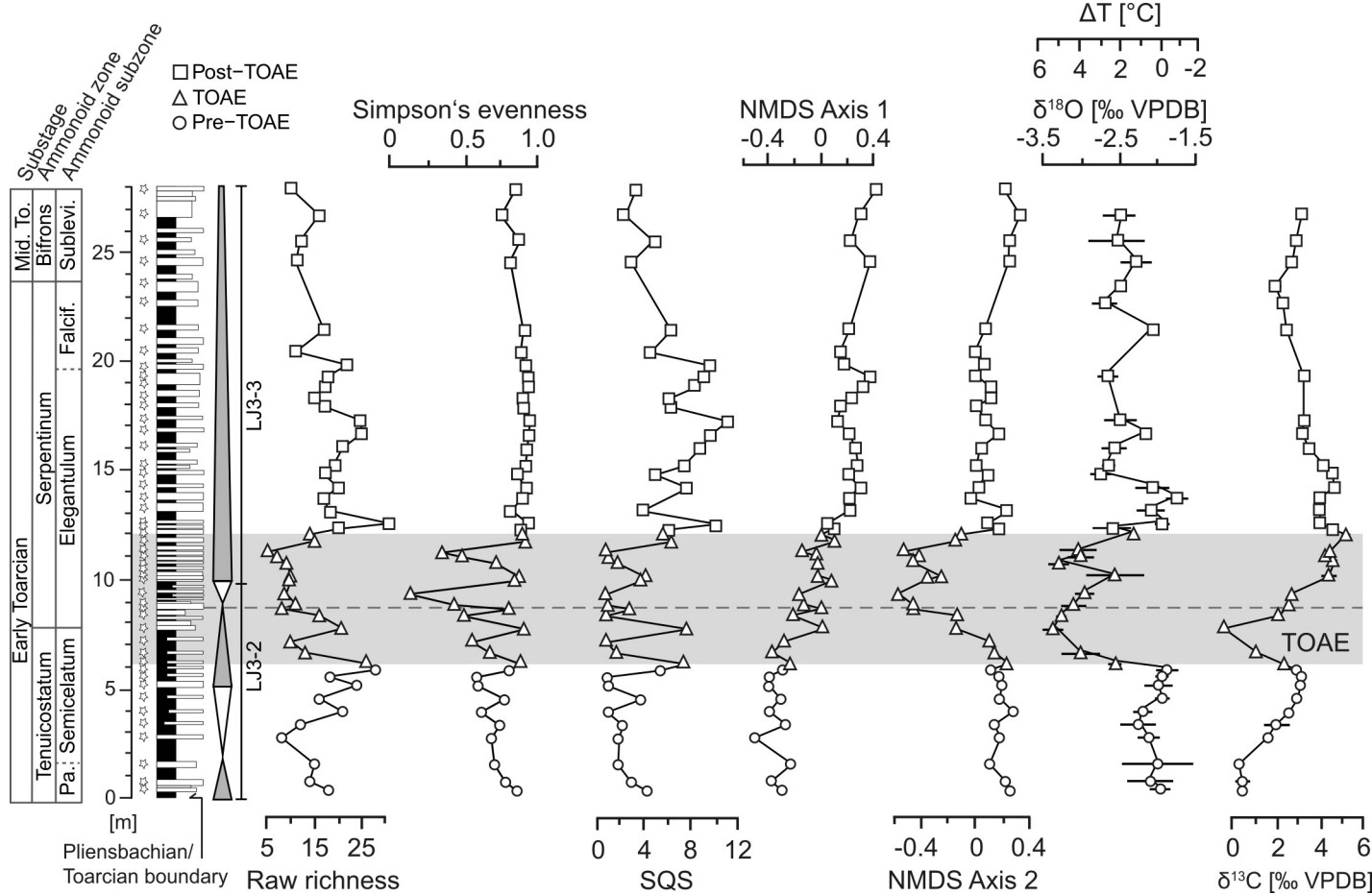

**Fig 3. Stratigraphy, transgressive-regressive cycles, geochemistry, and paleoecology at Barranco de la Cañada.** Biodiversity metrics (richness, evenness, and Shareholder Quorum Subsampling [SQS] diversity) are based on the taxonomic composition of faunal samples. Non-metric multidimensional scaling (NMDS) curves represent the values of axis 1 and axis 2 of the NMDS ordination of Fig 5A. SQS-diversity calculated after Alroy [56]. Third order transgressive-regressive cycles from refs. [15,53]. Isotope data from ref. [57]; the temperature change ΔT was calculated relative to the lowermost value recorded in the section using Brand's equation [58], following refs. [45,57]. Horizontal bars in the isotope time series represent the double standard error for each sample. For additional information, see caption of Fig 2.

temperatures were elevated by ca. 3.5˚C through the entire TOAE [57]. Although somewhat smaller compared to some previous studies (see above), this temperature change is considered reliable as it represents averages for pre-TOAE, TOAE and post-TOAE intervals from a large quantity (>400) of measurements with high stratigraphic resolution. Towards the end of the TOAE temperatures decreased but during the post-TOAE remained on average higher than during the pre-TOAE (Fig 3).

## Material and methods

We performed quantitative bed-by-bed sampling from carbonate beds, standardized by consistently collecting ~13 kg of bulk rock for each sample. We investigated the complete coverage of benthic macroinvertebrates, regardless of preservation quality. A total number of 3668 individuals belonging to 93 taxa was recorded, comprising 2078 bivalves (62 taxa), 1400 brachiopods (21 taxa), 152 gastropods (nine taxa) and 38 individuals of one coral species (Fig 2). Taxa were identified preferentially at species level, wherever preservation allowed. Specimens that could not reliably be identified at genus level were disregarded. This set of specimens has already been described and investigated for trends in body size in ref. [45].

We obtained oxygen and carbon isotopes from calcite shells of the best-preserved rhynchonellid brachiopods and oysters from each sample (see ref. [57] for distribution of samples). Sample extraction was performed with preparation needle or scalpel after sediment matrix, altered crusts and, in the case of brachiopods, the primary shell layer was removed. Sampling areas where the shell calcite visually appeared well preserved were targeted and analytical results further scrutinized by comparison to ultrastructure information for each sampled specimen and measurement of element/Ca ratios for each sample. The methods applied for the geochemical analyses are described in detail in ref. [57]. In short, samples of ca. 500 µg size were analysed using a Sercon 20–22 Gas Source isotope ratio mass spectrometer in continuous flow setup, with the reaction of carbonate with nominally anhydrous phosphoric acid under a He atmosphere taking place at 70˚C. Instrumental drift and biases were corrected with a two-point calibration using in-house standards CAR (Carrara Marble, $\delta^{13}C$ = +2.10 ‰; $\delta^{18}O$ = -1.93 ‰) and NCA (Namibian Carbonatite, $\delta^{13}C$ = -5.63 ‰; $\delta^{18}O$ = -21.90 ‰). These standards were previously calibrated against international standards, ensuring accuracy of the data. Precision of the measurements (2 s.d.) was found to be ± 0.07 ‰ for $\delta^{13}C$ and ± 0.15 ‰ for $\delta^{18}O$ based on 125 repeat analyses of CAR measured together with the fossil calcite samples.

Fossil preservation differs by taxonomical group: shells are well preserved in brachiopods, while in bivalves preservation is either as shell or moulds (both internal or external). Gastropods and corals occur generally as internal moulds. Because originally aragonitic taxa are preserved as moulds, the species composition and abundance distribution of the shelly macrobenthos apparently were not strongly influenced by diagenetic processes. Moreover, the absence of preferential shell orientations and lack of size sorting [45] suggest that shell transport was minimal, shells were buried within their original habitat, and faunal composition was not heavily biased by taphonomic processes.

The specimens collected in the field are deposited at the Museo de Ciencias Naturales de Universidad de Zaragoza, Spain [64] with inventory numbers MPZ 2019/415 –MPZ 2019/571, MPZ 2019/792 –MPZ 2019/889, MPZ 2019/920 –MPZ 2019/942 and MPZ 2019/1041. All necessary permits were obtained for the described study, which complied with all relevant regulations. Approval of the conduction of paleontological fieldwork was obtained from the Direccion General de Cultura y Patrimonio of Zaragoza (Gobierno de Aragon).

### Faunal analyses

Faunal analyses are based on a database of raw abundances of species standardized by rock volume and were performed in the R environment (v. 3.5.3, [65]). They address potential changes in taxonomic composition as well as in the ecological structure of benthic communities. Accordingly, we assigned each species to a specific mode of life (MOL)–a unique combination of feeding strategy, tiering and motility level reflecting the realized ecospace of a community [66,67]. MOL assignations (Table 1) are based on the published literature or were inferred from living relatives. Counts of species and MOLs were used to calculate the diversity indices (see below). Relative abundances (percentages) were used when applying ordination and clustering methods. In all analyses, levels with less than 17 specimens were excluded.

Our aim was to visualize to which degree pre-TOAE, TOAE and post-TOAE intervals differ in terms of taxonomical and ecological composition. Q-mode hierarchical clustering ('hclust' function of 'stats' package [65]) was based on a Bray-Curtis distance matrix ('vegdist' function in 'vegan' [68]), and Ward's clustering criterion [69] was implemented. The similarity profile test (SIMPROF) from the 'clustsig' package [70] was applied to determine the number of significant clusters [71].

**Table 1. Macrobenthic marine invertebrates recorded from the early Toarcian of Barranco de la Cañada, Spain, and their ecological attributes.**

| Fig 2[a] | Taxon | Motility | Tiering | Feeding |
|---|---|---|---|---|
| Bivalvia | | | | |
| 1 | *Gryphaea* (*Catinula*) cf. *crickleyensis* | Cem | Epi | Susp |
| 2 | *Pholadomya* (*Pholadomya*) *ambigua* | Fac-Mob(un) | D-inf | Susp |
| 3 | *Arcomya* sp. | Fac-Mob(un) | D-inf | Susp |
| 4 | *Arcomytilus pectinatus* | Bys | Epi | Susp |
| 5 | *Bositra buchi* | Fr(un) | Epi | Susp |
| 6 | *Isocyprina* sp. | Fac-Mob(un) | Shal-inf | Susp |
| 7 | *Lycettia lunularis* | Bys | Epi | Susp |
| 8 | *Pseudolimea* cf. *roemeri* | Bys | Epi | Susp |
| 9 | *Weyla* (*Weyla*) *pradoana* | Fr(un) | Epi | Susp |
| 10 | *Anisocardia* sp. | Fac-Mob(un) | Shal-inf | Susp |
| 11 | *Pronoella* (*Gythemon*) sp. | Fac-Mob(un) | Shal-inf | Susp |
| 12 | *Nicaniella voltzi* | Fac-Mob(un) | Shal-inf | Susp |
| 13 | *Harpax spinosa* | Cem | Epi | Susp |
| 14 | *Harpax auricola* | Cem | Epi | Susp |
| 15 | *Pseudopecten* (*Pseudopecten*) *dentatus* | Fac-Mob(un) | Epi | Susp |
| 16 | *Pleuromya aequistriata* | Fac-Mob(un) | D-inf | Susp |
| 17 | *Placunopsis radiata* | Cem | Epi | Susp |
| 18 | *Pinna* sp. | Bys | S-inf | Susp |
| 19 | *Girardotia* sp. | Fac-Mob(un) | D-inf | Susp |
| 20 | *Isognomon* sp. | Bys | Epi | Susp |
| 21 | *Harpax rapa* | Cem | Epi | Susp |
| 22 | *Chlamys* (*Chlamys*) *textoria* | Fac-Mob(at) | Epi | Susp |
| 23 | *Coelopis* (*Coelopis*) *similis* | Fac-Mob(un) | Shal-inf | Susp |
| 24 | *Parvamussium pumilum* | Fac-Mob(un) | Epi | Car |
| 25 | *Bakevellia* cf. *waltoni* | Bys | S-inf | Susp |
| 26 | *Corbulomima* cf. *obscura* | Fac-Mob(un) | Shal-inf | Susp |
| 27 | *Ctenostreon rugosum* | Fr(un) | Epi | Susp |
| 28 | *Liostrea* sp. | Cem | Epi | Susp |
| 29 | *Nanogyra* sp. | Cem | Epi | Susp |
| 30 | *Gervillella monotis* | Bys | S-inf | Susp |
| 31 | *Cucullaea* (*Idonearca*) *muensteri* | Bys | S-inf | Susp |
| 32 | *Myophorella* cf. *formosa* | Fac-Mob(un) | Shal-inf | Susp |
| 33 | *Cardinia concinna* | Fac-Mob(un) | Shal-inf | Susp |
| 34 | *Ceratomya concentrica* | Fac-Mob(un) | Shal-inf | Susp |
| 35 | *Camptonectes* (*Camptonectes*) *auritus* | Fac-Mob(at) | Epi | Susp |
| 36 | *Eopecten velatus* | Bys | Epi | Susp |
| 37 | *Gryphaea* (*Gryphaea*) cf. *dumortieri* | Fr(un) | Epi | Susp |
| 38 | *Spondylopecten* cf. *subspinosus* | Bys | Epi | Susp |
| 39 | *Protocardia* (*Protocardia*) *striatula* | Fac-Mob(un) | Shal-inf | Susp |
| 40 | *Myoconcha* sp. | Bys | S-inf | Susp |
| 41 | *Camptonectes subulatus* | Fac-Mob(at) | Epi | Susp |
| 42 | *Arcomytilus asper* | Bys | Epi | Susp |
| 43 | *Pholadomya* (*Pholadomya*) cf. *fidicula* | Fac-Mob(un) | D-inf | Susp |
| 44 | *Costigervillia* sp. | Bys | S-inf | Susp |
| 45 | *Plagiostoma punctatum* | Bys | Epi | Susp |
| 46 | *Homomya* sp. | Fac-Mob(un) | D-inf | Susp |

*(Continued)*

**Table 1.** (Continued)

| Fig 2[a] | Taxon | Motility | Tiering | Feeding |
|---|---|---|---|---|
| 47 | *Trigonia* (*Trigonia*) *similis* | Fac-Mob(un) | Shal-inf | Susp |
| 48 | *Entolium corneolum* | Fac-Mob(un) | Epi | Susp |
| 49 | *Pronoella* (*Pronoella*) *intermedia* | Fac-Mob(un) | Shal-inf | Susp |
| 50 | *Actinostreon* sp. | Cem | Epi | Susp |
| 51 | *Gryphaea* (*Bilobissa*) *sublobata* | Fr(un) | Epi | Susp |
| 52 | *Pinna* (*Pinna*) cf. *folium* | Bys | S-inf | Susp |
| 53 | *Pleuromya uniformis* | Fac-Mob(un) | D-inf | Susp |
| 54 | *Pseudolimea duplicata* | Bys | Epi | Susp |
| 55 | *Mactromya cardioides* | Fac-Mob(un) | Shal-inf | Susp |
| 56 | *Plagiostoma giganteum* | Fr(un) | Epi | Susp |
| 57 | *Modiolus* (*Modiolus*) *imbricatus* | Bys | S-inf | Susp |
| 58 | *Parallelodon hirsonensis* | Bys | S-inf | Susp |
| 59 | *Grammatodon* (*Grammatodon*) *concinnus* | Fac-Mob(at) | Epi | Susp |
| 60 | *Pholadomya* (*Pholadomya*) *reticulata* | Fac-Mob(un) | D-inf | Susp |
| 61 | *Inoperna sowerbiana* | Bys | S-inf | Susp |
| 62 | *Neocrassina* cf. *elegans* | Fac-Mob(un) | Shal-inf | Susp |
| Brachiopoda | | | | |
| 63 | *Gibbirhynchia cantabrica* | Ped | Epi | Susp |
| 64 | *Aulacothyris resupinata* | Ped | Epi | Susp |
| 65 | *Gibbirhynchia micra* | Ped | Epi | Susp |
| 66 | *Tetrarhynchia subconcinna* | Ped | Epi | Susp |
| 67 | *Lobothyris subpunctata* | Ped | Epi | Susp |
| 68 | *Lobothyris arcta* | Ped | Epi | Susp |
| 69 | *Liospiriferina falloti* | Ped | Epi | Susp |
| 70 | *Quadratirhynchia attenuata* | Ped | Epi | Susp |
| 71 | *Soaresirhynchia* aff. *flamandi* | Ped | Epi | Susp |
| 72 | *Soaresirhynchia bouchardi* | Ped | Epi | Susp |
| 73 | *Choffatirhynchia* aff. *paucicostatae* | Ped | Epi | Susp |
| 74 | *Gibbirhynchia* aff. *muirwoodae* | Ped | Epi | Susp |
| 75 | *Telothyris jauberti* | Ped | Epi | Susp |
| 76 | *Lingularia* sp. | Fac-Mob(at) | Shal-inf | Susp |
| 77 | *Choffatirhynchia vasconcellosi* | Ped | Epi | Susp |
| 78 | *Homoeorhynchia batalleri* | Ped | Epi | Susp |
| 79 | *Sphaeroidothyris dubari* | Ped | Epi | Susp |
| 80 | *Lobothyris hispanica* | Ped | Epi | Susp |
| 81 | *Homoeorhynchia meridionalis* | Ped | Epi | Susp |
| 82 | *Sphaeroidothyris perfida* | Ped | Epi | Susp |
| 83 | *Telothyris pyrenaica* | Ped | Epi | Susp |
| Gastropoda | | | | |
| 84 | *Lamelliphorus heliacus* | Reg-Mov | Epi | Her |
| 85 | *Neritopsis* sp. | Reg-Mov | Epi | Her |
| 86 | *Oonia* sp. | Reg-Mov | Epi | Her |
| 87 | *Katosira* sp. A | Reg-Mov | Epi | Her |
| 88 | *Katosira* sp. B | Reg-Mov | Epi | Her |
| 89 | *Cylindrobullina* sp. | Reg-Mov | Epi | Her |
| 90 | *Pseudomelania* sp. | Reg-Mov | Epi | Her |
| 91 | *Ampullospira* sp. | Reg-Mov | Epi | Her |

(*Continued*)

**Table 1.** (Continued)

| Fig 2[a] | Taxon | Motility | Tiering | Feeding |
|---|---|---|---|---|
| 92 | *Pietteia* sp. | Reg-Mov | Shal-inf | Surf-dep |
| Coelenterata | | | | |
| 93 | *Montlivaltia* sp. | Fr(un) | Epi | Car |

Abbreviations: Motility Level, stationary categories: Ped = Pedically attached; Bys = Byssate; Cem = Cemented; Fr(un) = Unattached free-lying; Motility Level, mobile categories: Fac-Mob(un) = Unattached facultative mobile; Fac-Mob(at) = Attached facultative mobile; Reg-Mov = Regularly moving. Tiering: Epi = Epifaunal; S-inf = Semi-infaunal; Shal-inf = Shallow infaunal; D-inf = Deep infaunal. Feeding Mechanisms: Susp = Suspension feeder; Her = Herbivore/grazing; Car = Carnivore/microcarnivore; Surf-dep = Surface deposit feeder.

[a] Numbering of taxa as in Fig 2.

Non-metric Multidimensional Scaling (NMDS) on a Bray-Curtis distance matrix was performed with the 'metaMDS' function from 'vegan' [68], with three dimensions and 100 attempts to search for a stable ordination solution. The fit of the NMDS was assessed by the stress value: the larger the stress value, the less efficient is the NMDS transformation [72]. Transformations with stress values lower than 0.2 were considered acceptable [72]. This ordination method is efficient in detecting distinct groupings in multivariate space and makes no assumptions about the distribution of the variables (e.g., [73]). The non-parametric Analysis of Similarities test (ANOSIM; 'anosim' function in 'vegan' [68], with 999 permutations) was used to test whether there is a significant difference between the three intervals represented in the NMDS.

We estimated the ecological importance of each MOL by calculating the average percentages per sample to visualize the relative abundance changes. The non-parametric Kruskal-Wallis test ([74]; 'kruskal.test' function in 'stats' [65]) and the associated *post hoc* Dunn's test [75] ('dunnTest' function in 'FSA' [76])—with Bonferroni correction for multiple comparisons— were used to investigate differences in the median relative abundance between intervals. The *post hoc* test was performed only when the output of the Kruskal-Wallis test was significant. To avoid decreasing the statistical power of the tests, we refrained from performing these analyses in the cases a MOL accounted for insufficient ($< 10$) numbers of individuals per interval. The median values by interval were then calculated and used to estimate the difference in percentage of each MOL between intervals. This allowed us to illustrate the magnitude and direction of change of single MOLs. These same procedures were performed for the most abundant trait belonging to each of the three ecological variables (feeding, motility and tiering). For simplification, motility was categorized as either active or passive (i.e. all taxa with no ability to move).

We calculated diversity indices, which gave us different aspects of biodiversity change, to investigate the taxonomical and functional biodiversity trends in time:

1. Alroy's Shareholder Quorum Subsampling (SQS [56]; R code available at 'http://bio.mq. edu.au/~jalroy/SQS-3-3.R'). This method was developed to calculate how many species (and MOLs in the case of functional diversity) can be found in each sample given a certain "quorum" (i.e. the desired frequency coverage level) of the abundance distribution. This method corrects for changing abundance distributions, thus balancing the samples. To include as many samples as possible, a quorum of 0.8 (= 80% coverage) was used for the ecological analyses, and of 0.6 to calculate taxonomical diversity;

2. Simpson's evenness index calculated with the 'diversity' function in 'vegan' [68], method 'simpson'. This is based on the formula 1- D, where $D = \Sigma p^2_i$ and $p_i$ the proportional abundance of $i$th species;

3. Raw richness, i.e. the total count of species per sampled horizon, was calculated with the function 'specnumber' in 'vegan' [68]. This measure is already standardized by amount of bulk rock sampled in the field.

## Correlation of faunal and geochemical data

To establish links of faunal variables with temperature change and/or processes determining the carbon cycle, we performed correlation tests for each of the biodiversity time series (SQS-diversity, Simpson's evenness, richness) and the compositional gradients (NMDS scores for axis 1 and 2) with the $\delta^{18}O$ and $\delta^{13}C$ isotope time series using Generalized Least Squares regression (GLS). The advantage of this method is that it can include terms to account for temporal autocorrelation among samples. Rows with missing values in any of the time series to be correlated were deleted prior to correlation.

We performed several steps before the GLS proper ('gls' function in 'nlme' package [77]):

i.   The presence of any trend in time was investigated for each time series through a simple Ordinary Least Squares (OLS) regression (X ~ sampling level, where X is each faunal, ecological and geochemical time series) (results reported in S1 Table);

ii.  We checked for autocorrelation with the Durbin-Watson statistic ('durbinWatsonTest' function from 'car' package [78]) of the relationship X ~ $\delta^{18}O$ + $\delta^{13}C$, where X refers to biodiversity indices or NMDS axis scores (S2 Table);

iii. As some of the time series indicate non-stationarity (step i) and/or autocorrelation (step ii), we performed generalized differencing for the purpose of detrending [79] with the R script available at 'http://www.graemetlloyd.com/pubdata/functions_2.r';

iv.  We repeated step ii) to check that the model assumptions of stationarity and non-autocorrelation were now satisfied (results of the Durbin-Watson statistic not reported).

v.   We fitted an AutoRegressive Integrated Moving Average (ARIMA) process with the auto. arima function in the "forecast" R package [80,81] to the residuals of the differenced OLS regressions X ~ $\delta^{18}O$ + $\delta^{13}C$. Because it is possible that unit roots are present as a different facet of an ARIMA process, the residual series of the fitted ARIMA process was investigated with the 'ndiffs' function from the "forecast" package ([80,81], with 'kpss' and 'adf' stationarity tests and maximum fifth order differencing applied). Given the negative outcome, we proceeded with the last step;

vi.  The ARIMA fit was finally applied in the GLS regression to specify the expected correlation structure of the residuals.

## Results

Bivalves dominate the studied fauna in terms of both species and number of specimens, with brachiopods being the second most abundant group (Fig 2). Gastropods and corals are rare faunal components (Table 1). Bivalves are most common and diverse in the pre-TOAE, but their diversity decreases approaching the TOAE. Many taxa, despite being rare or absent in the TOAE, range through and are again found in the post-TOAE. In contrast, brachiopods experience the extinction of pre-TOAE taxa in the lowermost Serpentinum Zone and a complete

turnover leading to a new brachiopod fauna in the post-TOAE (Fig 2). The pre-TOAE and TOAE intervals are characterized by high abundances in just a few or only one species, whereas species abundance distributions are more even in the post-TOAE. Ecologically, two MOLs are dominant: free-lying epifaunal suspension feeders in the pre-TOAE, and pedically attached epifaunal suspension feeders in the TOAE and in the post-TOAE.

## Change in taxonomic composition across the TOAE

Overall, the raw number of species increases from interval to interval, with 49 species sampled in the pre-TOAE, 55 in the TOAE and 68 in the post-TOAE. Moderately high and relatively stable diversity characterizes the pre-TOAE (Fig 3). Evenness and SQS-diversity vary little during the pre-TOAE, while raw richness increased during the Semicelatum Subzone until the onset of the TOAE. The TOAE itself is characterized by generally low values of raw species richness and contains the samples with the lowest evenness values. Fluctuations in all biodiversity indices are more pronounced compared to the other intervals (Fig 3). A shift to relatively high values in all three diversity metrics took place during uppermost TOAE and earliest post-TOAE times. Diversity values remain high and, in the case of evenness, stable throughout the studied post-TOAE interval.

Concerning faunal composition in more detail, cluster analysis on the 46 sampled horizons yielded 12 faunal associations, each consisting of faunal samples with similar taxonomic composition and abundance distribution of species (Fig 4A; version with stratigraphic levels provided in the Supplement as S1A Fig). The pre-TOAE, TOAE, and post-TOAE intervals are generally well separated, which is confirmed by the NMDS ordination (Fig 5A) and NMDS axes scores (Fig 3) (ANOSIM: R = 0.701, $p$-value = 0.001). Whereas some overlap in NMDS space exists between pre-TOAE and TOAE intervals, the post-TOAE samples are distinct from those of both other intervals.

The pre-TOAE is taxonomically characterized by the dominance of the oyster *Gryphaea sublobata*, and secondarily by terebratulid brachiopods belonging to the genus *Lobothyris* (Fig 2). These taxa are still an important part of the fauna during the early stages of the TOAE, hence the similarity of some TOAE assemblages with those from the pre-TOAE (Figs 2 and 5A). After the typical pre-TOAE brachiopods disappeared during the lower part of the TOAE (Fig 2), the rhynchonellid brachiopod *Soaresirhynchia bouchardi* becomes the dominant taxon in the upper part of the TOAE. The levels strongly dominated by *S. bouchardi* correspond to the evenness minima in Fig 3. The most common bivalves of the TOAE interval are *Entolium corneolum* and *Pinna* cf. *folium*. Brachiopods are still an important faunal component in the post-TOAE interval (Fig 2) and are represented mostly by terebratulid species of the genus *Telothyris*. As for the bivalves, *Grammatodon concinnus*, *E. corneolum* and *Protocardia striatula* are the most common species.

In terms of absolute abundances, numbers of individuals are high during the pre-TOAE and decline with the onset of the TOAE, reaching lowest values in the upper part of the TOAE (Fig 6). As an exception, two TOAE samples exhibit particularly high abundances, owing to high abundances of the rhynchonellid brachiopod *Soaresirhynchia bouchardi* (see below). Abundances again increase in the post-TOAE but mostly stayed below pre-TOAE levels.

## Changes in ecological composition

The trajectories of functional diversity (Fig 6) resemble those based on taxonomic composition. Specifically, the TOAE is distinct from the other two intervals by exhibiting marked fluctuations in all curves. As was the case for taxonomic composition, pre- and post-TOAE intervals also differ in ecological composition (Figs 4B and 5B) (ANOSIM: R = 0.488, $p$-

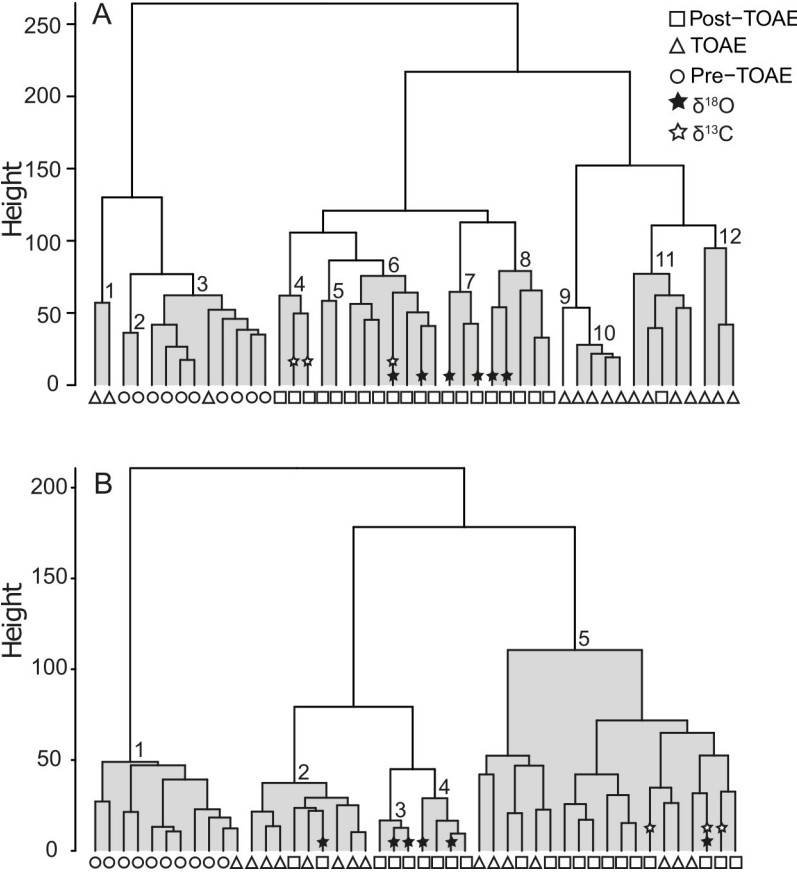

**Fig 4. Cluster analyses of the faunal samples at Barranco de la Cañada.** Results plotted according to taxonomical composition using species as variables (A) and ecological composition using modes of life as variables (B). The identified associations are shaded in grey and each sample is coded by interval (pre-TOAE, TOAE and post-TOAE). The stars mark the levels in the post-TOAE where the recorded isotope values (both $\delta^{18}$O and $\delta^{13}$C) are within their pre-TOAE ranges.

value = 0.001). The ecologically fairly homogeneous pre-TOAE (Fig 4B) is dominated by free-lying (ca. 50%) and pedically attached (ca. 20%) epifaunal suspension feeders (Fig 7), represented respectively by ostreid bivalves and terebratulid brachiopods. The TOAE marks a major decline of epifaunal free-lying suspension feeders and an increase in pedically attached suspension feeders (Fig 7). The latter, represented by two species of the brachiopod genus *Lobothyris* in the early part of the TOAE and *S. bouchardi* after the brachiopod turnover, are dominant in the TOAE (> 40%), while free-lying epifaunal individuals are reduced to < 10%.

In ecological composition, TOAE samples often overlap with post-TOAE samples (Figs 4B and 5B) and the dominant MOL in the TOAE is still the most important in the post-TOAE (on average ca 40% of the post-TOAE fauna are pedically attached brachiopods). No MOL present in the pre-TOAE is lost after the TOAE.

By calculating the change in the percentage of each MOL among intervals (Fig 8), we confirm that the most substantial changes take place within two MOLs, i.e. the free-lying and the pedically attached suspension feeders. In particular, changes in these two MOLs are statistically significant for the pre-TOAE interval when compared to both other intervals (Table 2). Most of the change occurred in the early phase of the TOAE, whereas subsequent changes during the post-TOAE are relatively minor in extent (Figs 7 and 8).

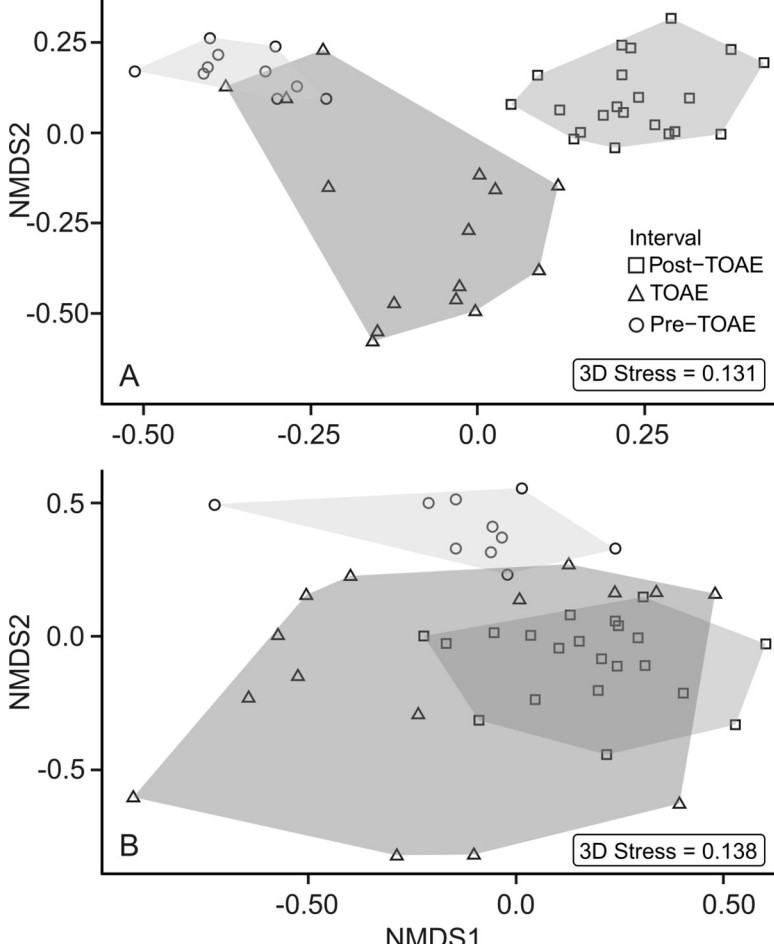

**Fig 5. Non-metric multidimensional scaling ordination (NMDS) of faunal samples at Barranco de la Cañada.**
Results plotted by taxonomical (A) and ecological (B) composition. The symbols and the shading of the convex-hull polygons are coded by interval (pre-TOAE, TOAE and post-TOAE).

When we considered motility, tiering, and feeding mechanism separately to pinpoint which traits within a MOL are most affected (Fig 9), we find that most change is located within taxa with passive motility. Changes in the relative abundance of epifaunal taxa and of suspension feeders are fairly small and significant only for suspension feeders (Table 2).

## Correlation of faunal data with $\delta^{13}C$ and $\delta^{18}O$ values

All taxonomically and functionally based biodiversity indices are significantly correlated with the $\delta^{18}O$ isotope time series whereas correlations with $\delta^{13}C$ values are all not significant (Table 3). Furthermore, the abundance distribution of MOLs is strongly tied to $\delta^{18}O$ values, as is evident from significant correlations with both NMDS axes scores (Table 3). These results suggest an important role of temperature change in the observed faunal patterns. It should be kept in mind that reconstructing the carbon cycle from the $\delta^{13}C$ record is much more complex than a comparatively simple conversion of $\delta^{18}O$ values to temperature estimates. Therefore, we cannot exclude that carbon cycle-relevant processes did have some impact on communities but their expression in $\delta^{13}C$ values is non-unique and only one of the contributing factors in shaping the observed $\delta^{13}C$ signal.

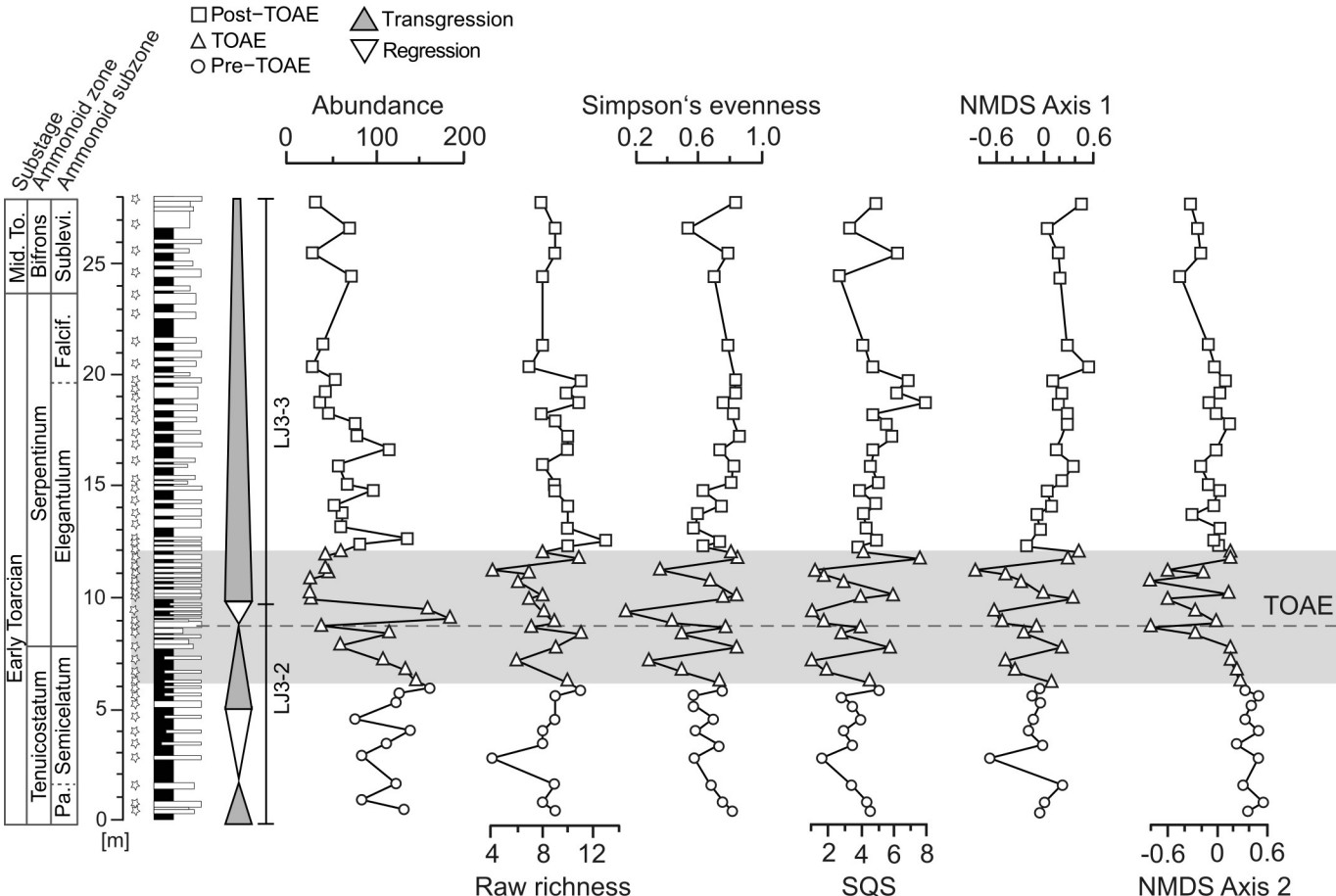

**Fig 6. Paleoecology of faunal samples at Barranco de la Cañada based on ecological composition.** Functional diversity indices (richness, evenness, and SQS-diversity) and NMDS axis scores of Fig 5B were calculated using modes of life as variables. Third order transgressive-regressive cycles from refs. [15,53]. The abundance of samples is shown as their respective number of individuals. For additional information, see captions of Figs 2 and 3.

## Discussion

### The role of warming versus other environmental stressors

We document substantial faunal changes across the studied early Toarcian interval that are correlated with changes in water temperature. For another locality of the Iberian Range (Castrovido), Danise et al. [9] found significant correlations of δ[18]O values with the taxonomic composition of Pliensbachian–Toarcian macrobenthic communities and bivalve subcommunities. Although community composition in our analysis was the only paleoecological variable that did not significantly correlate with the δ[18]O time series (Table 3), this finding is an independent indicator that temperature variations have influenced the composition of early Toarcian communities. Also, unlike our study, correlations between δ[18]O values and diversity indices (richness, evenness) were not significant for Castrovido and another Spanish locality (Sierra Palomera; [9]). This lack of correlations might be influenced by their relatively low number of observations for the TOAE interval and the stratigraphic clustering of brachiopod-derived isotope values.

Apart from heat stress during the TOAE hyperthermal, the observed faunal patterns could also be caused by other or additional stressors related to climate warming. These include seawater dysoxia, reduced salinity, ocean acidification, changes in primary productivity, and

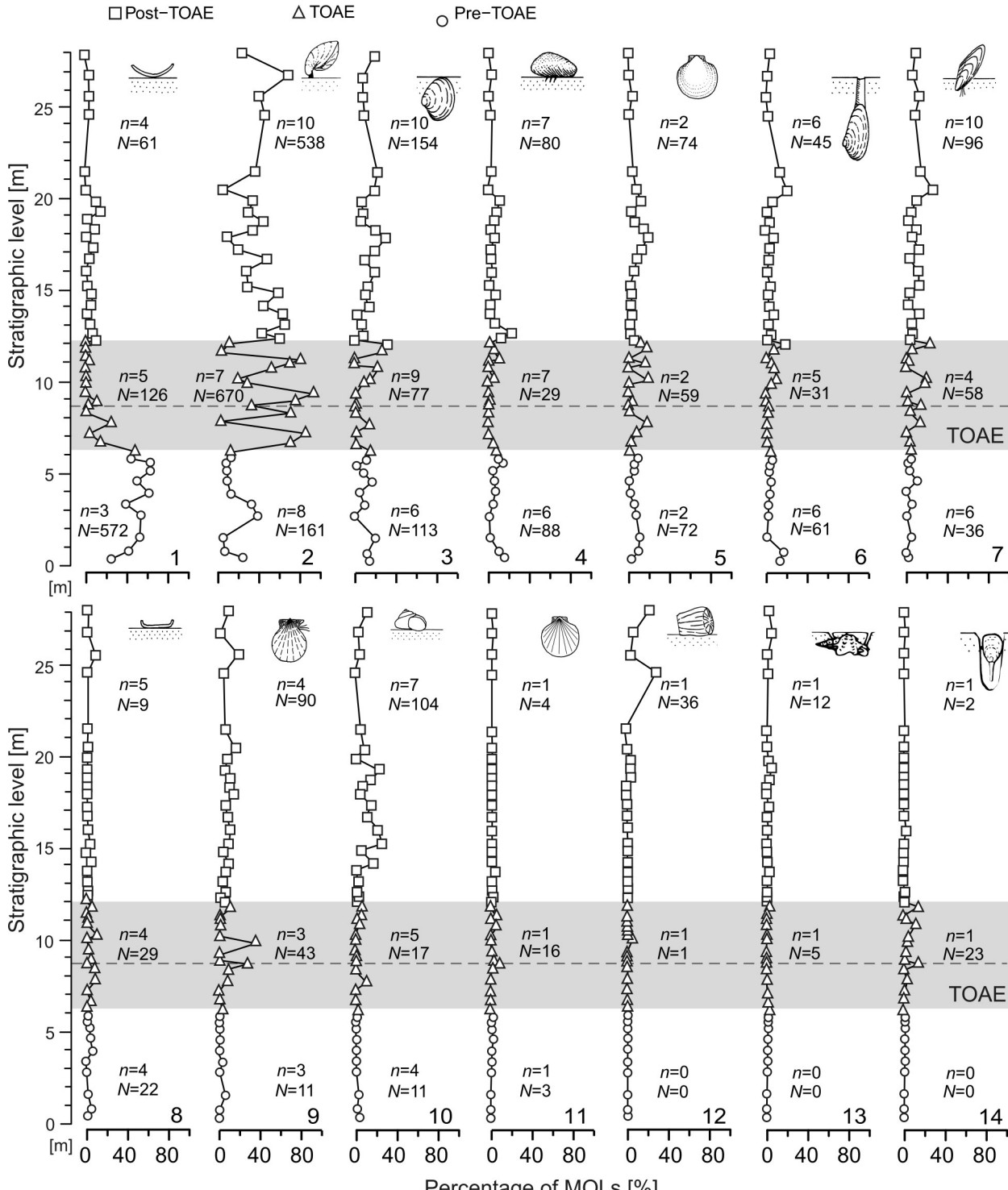

**Fig 7. Relative abundance of each mode of life (MOL).** The shaded area represents the TOAE interval, and the dashed horizontal line marks the level with the last appearances of the pre-TOAE brachiopod fauna. Legend: "$n$" = number of species; "$N$" = number of individuals. MOL codings: 1 = Unattached free-lying epifaunal suspension feeders; 2 = Pedically attached epifaunal suspension feeders; 3 = Unattached facultatively mobile shallow infaunal suspension feeders; 4 = Byssate epifaunal suspension feeders; 5 = Unattached facultatively mobile epifaunal suspension feeders; 6 = Unattached facultatively mobile deep infaunal suspension feeders; 7 = Byssate semi-infaunal suspension feeders; 8 = Cemented epifaunal suspension feeders; 9 = Attached facultatively mobile epifaunal suspension feeders; 10 = regularly moving epifaunal herbivore; 11 = Unattached facultatively mobile epifaunal carnivore; 12 = Unattached free-lying epifaunal carnivore, 13 = Regularly moving shallow infaunal surface deposit feeders; 14 = Attached facultatively mobile shallow infaunal suspension feeders.

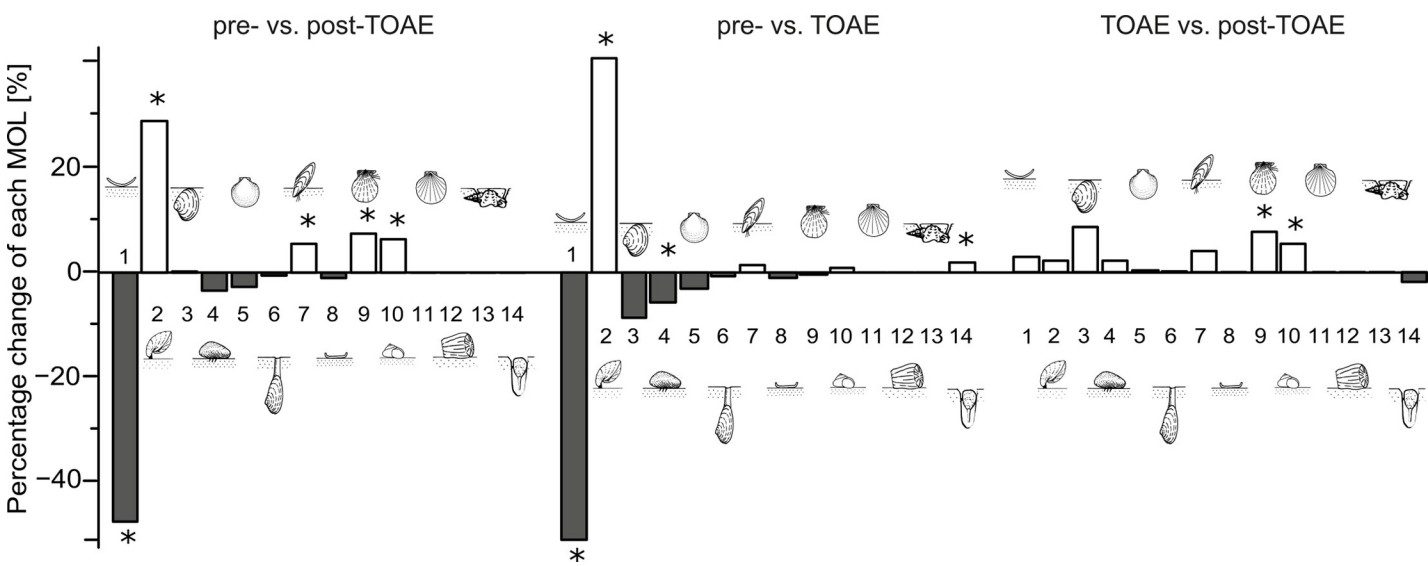

**Fig 8. Relative change of each mode of life (MOL) among time intervals.** Comparisons are based on medians. White and dark grey bars represent positive and negative change, respectively. The symbol (*) denotes significant *p*-values from the Kruskal-Wallis and Dunn's tests (Table 2). For MOL codings, see caption of Fig 7.

changes in habitat via sea level change. For the studied site, we discussed these factors in detail in two recent articles [45,57] and concluded that changes in temperature are the best

**Table 2. Results of the non-parametric Kruskal-Wallis test on medians and post-hoc Dunn's test.**

| MOL/Trait | Kruskal-Wallis test | | Dunn's test | | | | | |
| | | | pre-TOAE vs. TOAE | | pre-TOAE vs. post-TOAE | | TOAE vs. post-TOAE | |
| | $\chi^2$ | *p*-value | Z | *p*-value | Z | *p*-value | Z | *p*-value |
|---|---|---|---|---|---|---|---|---|
| 1 | 22.919 | **<0.001** | 4.520 | **<0.001** | 4.096 | **<0.001** | -0.803 | 1 |
| 2 | 9.423 | **0.009** | -2.877 | **0.012** | -2.658 | **0.024** | 0.453 | 1 |
| 3 | 2.144 | 0.342 | / | / | / | / | / | / |
| 4 | 7.118 | **0.029** | 2.636 | **0.025** | 1.363 | 0.518 | -1.634 | 0.307 |
| 5 | 0.844 | 0.656 | / | / | / | / | / | / |
| 6 | 1.177 | 0.555 | / | / | / | / | / | / |
| 7 | 7.690 | **0.021** | -1.381 | 0.502 | -2.733 | **0.019** | -1.438 | 0.451 |
| 8 | 4.872 | 0.088 | / | / | / | / | / | / |
| 9 | 14.656 | **0.001** | -1.416 | 0.470 | -3.643 | **0.001** | -2.430 | **0.045** |
| 10 | 15.892 | **<0.001** | -0.681 | 1 | -3.460 | **0.002** | -3.110 | **0.006** |
| 11 | | | | | | | | |
| 12 | | | | | | | | |
| 13 | | | | | | | | |
| 14 | 20.113 | **<0.001** | -3.734 | **0.001** | -0.466 | 1 | 3.979 | **<0.001** |
| Pas | 4.994 | 0.082 | / | / | / | / | / | / |
| Epi | 1.830 | 0.401 | / | / | / | / | / | / |
| Susp | 22.185 | **<0.001** | 1.870 | **<0.001** | 4.523 | **<0.001** | 2.882 | **<0.001** |

Dunn's post-hoc test was performed in case of a significant result from the Kruskal-Wallis test to investigate which pairwise-comparison by interval is significant, otherwise the symbol "/" was used. The reported *p*-values for the post-hoc test are adjusted after applying the Bonferroni correction for multiple comparisons. Analyses were performed for each MOL and single traits. Statistically significant values (*p* < 0.05) are in bold. $\chi^2$ values based on two degrees of freedom. For coding of MOLs (1–14) see caption of Fig 7. Pas = Passive; Epi = Epifauna; Susp = Suspension feeders.

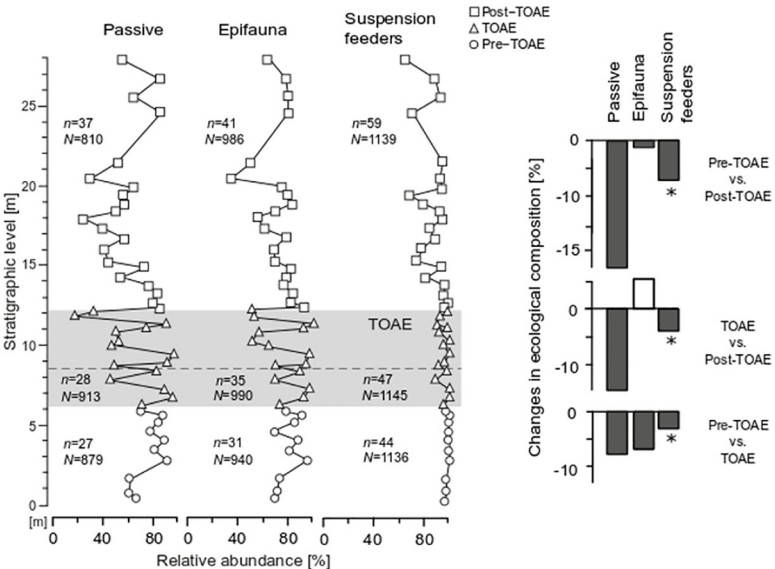

**Fig 9. Relative abundance and relative change of the most important ecological traits.** Each depicted trait represents the most abundant category in each of the three ecological characters that define a MOL (motility, tiering and feeding mechanism). The change among intervals is calculated from the median abundance of all sample means for each interval. The symbol (*) denotes significant *p*-values from the Kruskal-Wallis and Dunn's tests (Table 2). White and dark grey bars represent positive and negative change, respectively. "*n*" = number of species, "*N*" = number of individuals per interval.

supported and most plausible factor. In brief, the well bioturbated and shell-rich TOAE sediments preclude oxygen-depleted bottom waters at Barranco de la Cañada. This is quite unlike more northern areas where black shales are common, and faunal change is related to ocean deoxygenation and changes in primary productivity (e.g., [28,29,82]). Freshening of seawater

**Table 3. Generalized Least Squares (GLS) correlations of geochemical proxy data with paleoecological variables.**

| | | *p*-value | | Coefficients | | Standard error | |
|---|---|---|---|---|---|---|---|
| | ARIMA (p,d,q) | $\delta^{18}O$ | $\delta^{13}C$ | $\delta^{18}O$ | $\delta^{13}C$ | $\delta^{18}O$ | $\delta^{13}C$ |
| **A Taxonomic composition** | | | | | | | |
| SQS | (0,0,0) | **0.047** | 0.598 | 2.595 | -0.312 | 1.258 | 0.585 |
| Simpson's Evenness | (0,0,0) | **0.014** | 0.535 | 0.212 | 0.024 | 0.082 | 0.0378 |
| Richness | (0,0,0) | **0.036** | 0.797 | 5.839 | -0.320 | 2.659 | 1.236 |
| NMDS1 | (1,0,2) | 0.915 | 0.428 | -0.006 | -0.018 | 0.054 | 0.022 |
| NMDS2 | (0,0,0) | 0.290 | 0.336 | 0.085 | 0.036 | 0.079 | 0.0367 |
| **B Ecological composition** | | | | | | | |
| SQS | (0,0,0) | **0.024** | 0.842 | 1.599 | 0.063 | 0.673 | 0.313 |
| Simpson Evenness | (0,0,0) | **0.012** | 0.703 | 0.203 | 0.014 | 0.076 | 0.035 |
| Richness | (0,0,0) | **0.032** | 0.890 | 1.791 | 0.052 | 0.800 | 0.372 |
| NMDS1 | (0,0,0) | **0.017** | 0.791 | 0.379 | -0.019 | 0.150 | 0.070 |
| NMDS2 | (0,0,0) | **0.009** | 0.869 | 0.328 | 0.009 | 0.118 | 0.055 |

Paleoecological variable represented by SQS-diversity, evenness, richness, and NMDS axis scores. The geochemical proxy data used in this study are $\delta^{18}O$ and $\delta^{13}C$ values obtained from brachiopod and oyster shells. The correlations were performed for faunal assemblages characterized by taxonomic composition (A) and by ecological composition (B) on differenced time series. Statistically significant values ($p < 0.05$) are in bold. Abbreviations of the parameters of the ARIMA process fitted to the GLS model: p = number of time lags; d = degree of differencing; q = order of moving average.

is incompatible with paleo-oceanographic modelling and with faunal composition (stenohaline ammonites, crinoids, and rhynchonelliform brachiopods are continuously present), whereas evidence of freshening exists for central and northern Europe [19,83], making salinity a stressor of regional importance. This geographic variability in salinity and oxygen regimes shows that, on larger spatial scales, multiple mechanisms can be operating simultaneously with different relative contributions in different parts of the ocean. Ocean acidification is more difficult to assess but is not supported by faunal patterns at Barranco de la Cañada, where brachiopods, which might be more tolerant against lowered pH than bivalves [84,85], are more strongly affected ([45], this study). Likewise, the selective extinction of brachiopods may argue against low productivity because their typically low metabolic rates and their capability to feed on both particulate and dissolved organic matter should buffer them against reduced food supply [86,87]. Finally, patterns of faunal change may also arise from facies change owing to fluctuations in sea level, and Danise et al. [9] showed the clustering of last occurrences of early Toarcian species in the Iberian Basin at flooding surfaces. At Barranco de la Cañada, facies changes are fairly subtle and we sampled largely homogeneous lithologies (marly wackestones to floatstones) with just one sample being from a rudstone (S1 Fig). Overall, the depositional environment stayed in a low-energy, mid-ramp position. Also, diversity trends (both taxonomical and ecological) are not consistent in equivalent stages of transgressive-regressive cycles. This is for example evident in Fig 3 where we would expect congruent diversity trends in each transgressive-regressive cycle if diversity fluctuations were primarily driven by fluctuations in sea level, which is not the case.

## Timing of faunal change

Changes in multiple community attributes (Figs 3 and 6) took place at the onset of the TOAE, synchronous with an increase in water temperatures. Within the first samples of the TOAE, taxonomical and functional diversity indices decrease and enter into a mode of strongly fluctuating values. Similarly, the two most important MOLs show drastic abundance changes right at the onset of the event (Fig 7). This suggests that in terms of ecological organisation the pre-existing ecosystem was not resistant to the geologically rapid temperature rise. By contrast, taxonomic composition shifted more gradually (see NMDS scores in Fig 3), and the main change happened around the transition between the early and the late phase of the TOAE, coinciding with the final disappearance of the pre-existing brachiopod fauna and the appearance of *Soaresirhynchia* assemblages. Similar patterns at the local scale are also known from modern ecosystems where–on much shorter time scales–anthropogenic disturbances predominantly decreased abundance and species richness [88]. In contrast to our findings, however, the meta-analysis of Murphy & Romanuk [88] showed that the effects of temperature increase on modern aquatic ectotherms were insignificant. This difference could be due to the magnitude and prolonged duration of raised temperatures in our study system or because increases in temperature need not be deleterious as long as organisms shift towards or stay close to their thermal optima.

Our observation of functional shifts at the onset of the TOAE is consistent with the timing of functional changes in dysoxic to anoxic settings [28,29,31,89]. However, apart from functional change, these dysoxic–anoxic environments also exhibit compositional shifts at the onset of the TOAE negative carbon isotopic excursion. This contrasts with our present findings from an oxygenated environment, where compositional changes shifted into the early phase of the TOAE. This discrepancy could suggest that the combination of warming and dysoxia in oxygen-depleted environments led to earlier, more abrupt shifts comprising both community composition and structure.

## Ecological state shift during hyperthermal conditions

The TOAE interval clearly differs from the intervals before and after in several faunal characteristics that are indicative of sustained stressful and ecologically unstable conditions in the benthic marine ecosystem. First, the species composition changed with the complete loss of the pre-TOAE brachiopod fauna in the first half of the TOAE. At a wider geographic scale, none of the pre-TOAE articulate brachiopod species of the Iberian platform system survived this episode [40]. In the second half of the TOAE, *Soaresirhynchia*-dominated assemblages appeared and prevailed, whereas most bivalve species disappeared or became rare. The dominance of *Soaresirhynchia* in the late phase of the TOAE is known from other basins in the Mediterranean [9,15–16,40,44,90,91]. *S. bouchardi* is a small-sized species with high morphological variability, geographically widespread (ranging from North Africa to England [40]), and occurs in high numbers. This taxon was successful especially during the rising limb of the TOAE negative carbon isotope excursion where it thrived despite persisting high water temperatures. Its presumably slow growth and low metabolic rates, as suggested by the geochemical composition and low organic matter content of the shell [57], probably put it at an advantage in a stressed environment over other species with higher energy demands [9,57]. Also, the associated bivalve *Parvamussium pumilum* and the lingulid brachiopod *Lingularia* sp. have been interpreted as opportunistic taxa capable of surviving in stressed environments [92,93], making the second half of the TOAE an interval dominated by opportunists.

Second, the abundance of specimens in samples (which is standardized by rock volume) is on average lower during the TOAE than in the pre-TOAE. Combined with the observation from the same samples that larger-sized species selectively become less abundant during the TOAE [45], this suggests an overall decline of macrobenthic biomass in the TOAE. Notable exceptions are two samples with proliferations of the opportunistic *S. bouchardi*.

Third, the shift from diverse pre-TOAE assemblages to the paucispecific assemblages of the TOAE is paralleled by distinct oscillations of both taxonomic and functional biodiversity indices during the TOAE (Figs 3 and 6). Along with pronounced fluctuations in MOL-based NMDS scores, in the proportions of passive and epifaunal individuals, and with high-amplitude excursions in the abundance of specimens (Figs 6 and 9), this points to ecological instability during the TOAE.

## Recovery from the TOAE hyperthermal

The latest phase of the TOAE is characterized by a swift recovery of both taxonomical and ecological biodiversity. Such a geologically quick recovery contrasts with other areas where the TOAE is characterized by black shales (e.g., [28,29,31,89]) and has been related to the more favourable conditions of Tethyan areas where anoxia and dysoxia did not develop [9]. The return to high diversity values in the latest TOAE is synchronous with the shift to more positive $\delta^{18}O$ values, with cooler temperatures indicating the end of the hyperthermal interval. We interpret this synchronicity of trends in temperature and biodiversity at the beginning and the end of the TOAE hyperthermal as evidence for an important role of warming in diversity loss and of cooling in the subsequent recovery of diversity. The speed of diversity recovery as soon as temperatures decline suggests that benthic diversity was resilient, albeit taxonomic composition had changed. Subsequent stable diversity in the post-TOAE interval indicates a return to ecosystem stability (Figs 3 and 6).

Comparing pre- and post-TOAE assemblages, we found that no MOL present in the pre-TOAE was lost after the TOAE. Yet, ordinations and cluster analyses (Figs 4 and 5) show that pre- and post-TOAE intervals differ taxonomically and ecologically from each other. Taxonomic disparity can readily be explained by the strong turnover in brachiopods and marked reductions in the abundance of the oyster *Gryphaea sublobata* that had dominated the pre-

TOAE interval. In the face of extinction, full recovery of taxonomic composition remains impossible. Yet, functional composition can be restored in the absence of taxonomic compositional recovery because different species can be functionally the same [26]. Post-TOAE temperatures were mostly higher compared to the pre-TOAE ([57], Fig 3). To test whether this difference in temperature explains the difference in ecological composition, we compared only those post-TOAE samples with pre-TOAE assemblages where $\delta^{18}O$ values stayed within their pre-TOAE range (see selected samples in Fig 4 and S1 Fig). Even these selected post-TOAE assemblages have a different ecological composition than before which shows that communities did not return to their previous state despite similar inferred temperatures.

## Comparison with modern ecosystems and projected warming scenarios

The early Toarcian sedimentary rocks of the Barranco de la Cañada section were deposited in a shallow to mid shelf/ramp environment at low, subtropical paleolatitudes. The dominance of bivalves and brachiopods in marine, shelly, macrobenthic associations in these level bottom ecosystems is common during the early Jurassic whereas the higher level taxonomic composition has changed to bivalve-gastropod associations in the late Cenozoic [94]. Although locally common in shallow waters, the distribution of living brachiopods reflects the continuation of a post-Permian trend in brachiopod retreat to offshore habitats, and today they typically are species of the deeper continental shelf and upper slope [86]. Because brachiopods in our study were more severely hit by temperature rise than bivalves one might expect that, everything else being equal, modern shallow-to-mid shelf assemblages are possibly more resistant to warming than those of the early Jurassic.

In terms of absolute temperatures, using the brachiopod oxygen isotope thermometer of Brand et al. [58], we estimated average pre-TOAE water temperatures at the seafloor of ca. 21˚C followed by a geologically rapid mean temperature increase during the TOAE of ca. 3.5˚C [57]. Climate change predictions until the end of the century range from a low greenhouse gas emission scenario with a projected mean of 0.73˚C warming of global mean sea surface temperature to a high emission scenario with mean warming of 2.58˚C relative to pre-industrial levels [95]. In this respect, the amount of early Toarcian warming inferred in our analysis [57] is beyond the range of ocean warming scenarios for the near future. Yet, this does not imply that the local ecological effects of future warming are less severe than those established for the early Toarcian. These temperature estimates are not equivalent because they represent local point estimates (Barranco de la Cañada) versus global mean sea surface temperature (in ref. [95]). Furthermore, the rate of projected warming is likely much higher than during the TOAE, making adaptations to warming oceans more difficult. Several other factors add to the uncertainties in assessing the consequences of current and projected climate change for marine ecosystems from deep-time warming analogs as recorded by fossils. These include the time-averaged nature of fossil assemblages, the complex interactions with other warming-related stressors which may vary geographically, differences in the thermal structure of communities, and differences in the starting state of the climate system [96–98]. Despite these complexities and uncertainties we tentatively consider the identified faunal changes across the TOAE hyperthermal–taxonomic and ecological reorganisations; declines in diversity, abundance, and biomass; long-term ecological instability with predominance of opportunistic species; and reductions in community size structure (see ref. [45])–to also be plausible threats to present-day shallow-water benthic communities.

## Conclusions

High-resolution early Toarcian faunal and geochemical data from marine mid-ramp environments at Barranco de la Cañada provide statistical support for a relationship between bottom

water temperatures and taxonomical and ecological changes in benthic macroinvertebrate assemblages. This result provides strong support of the regional importance of warming as a proximate cause of the TOAE-related biotic crisis. Because dysoxia–anoxia prevailed in many other regions during the TOAE, our study alongside others suggests that multiple mechanisms can be operating simultaneously with different relative contributions in different parts of the ocean.

The local faunal assemblages were reorganized taxonomically and ecologically across the TOAE. This suggests that the local ecosystem could not withstand the disturbance, heat stress exceeded the communities' resistance capacity, and an environmental threshold was crossed leading to a non-transient change with a new distribution of species-specific traits in the TOAE aftermath. Brachiopods were most severely hit, as they experienced species extinctions in the lower part of the TOAE followed by the proliferation of an opportunistic species and a complete species turnover in the post-TOAE interval. Molluscs, on the contrary, mostly ranged through the event, hinting at rather different responses to environmental stress among taxonomical groups. Most of the ecological changes were focused at the onset of the TOAE hyperthermal and comprise declines in multiple diversity metrics, abundance, and biomass. During the TOAE, the studied marine communities were ecologically unstable, as evidenced by oscillatory dynamics in most ecological variables, whereas stable and diverse post-TOAE faunal assemblages were established simultaneously with decreasing water temperatures at the end of the TOAE. This suggests geologically rapid functional recovery to an undisturbed state, albeit with different frequency distributions of modes of life than before the event.

This confirms the need for a better understanding of ecosystem and organism responses under heat stress. Investigations of deep-time episodes of climate change like the TOAE can provide insights how marine communities might ultimately respond in the face of the present global warming and to changes in future climatic regimes.

## Supporting information

**S1 Fig. Cluster analyses of the sampled fauna at Barranco de la Cañada.** Clusters from Fig 4 plotted showing the stratigraphic level and lithology of each sample in meters above the Pliensbachian-Toarcian boundary. Results plotted by taxonomical (A) and ecological (B) composition. The identified associations are numbered and shaded in grey and each sample is coded by interval (pre-TOAE, TOAE and post-TOAE). The stars mark the levels in the post-TOAE where the recorded isotope values (both $\delta^{18}$O and $\delta^{13}$C) are within pre-TOAE ranges. (TIF)

**S1 Table. Results of the Ordinary Least Squares (OLS) correlations of faunal variables and of geochemical proxy data against time (sampling level).** Faunal variables are SQS-diversity, evenness, richness, and NMDS axis scores, whereas the geochemical proxy data are $\delta^{18}$O and $\delta^{13}$C values. Results are presented for faunal assemblages characterized by both taxonomic and ecological composition respectively and are based on the original time series. Statistically significant values ($p < 0.05$) are in bold. (DOCX)

**S2 Table. Durbin-Watson (D-W) statistics on Ordinary Least Squares (OLS) correlations of faunal variables with geochemical proxy data.** The OLS correlations are in the form X~ $\delta^{18}$O + $\delta^{13}$C, where X is the respective diversity index or NMDS axis score. Results are presented separately for faunal assemblages characterized by taxonomic and by ecological composition based on the original time series. Statistically significant values ($p < 0.05$) are in bold. (DOCX)

## Acknowledgments

We warmly thank Tina Klein and Friedrich Lucassen (Center for Marine Environmental Sciences, University of Bremen) for joint work and assistance in the field. We are grateful to Juan Carlos Garcia and the Direccion General de Cultura y Patrimonio (Gobierno de Aragon, Zaragoza) for authorizing our fieldwork, and to Jose Ignacio Canudo (Universidad de Zaragoza) for the loan of specimens. We thank Carl J. Reddin and David Lazarus (Museum für Naturkunde Berlin) for their suggestions for improvements of the manuscript and the journal reviewers Uwe Brand and Daniel Killam for their constructive reviews. This study is part of the Research Unit TERSANE (FOR 2332: Temperature-related Stressors as a Unifying Principle in Ancient Extinctions) and a contribution to the IGCP-655 (IUGS-UNESCO: Toarcian Oceanic Anoxic Event: Impact on Marine Carbon Cycle and Ecosystems).

## Author Contributions

**Conceptualization:** Veronica Piazza, Martin Aberhan.

**Data curation:** Veronica Piazza.

**Formal analysis:** Veronica Piazza.

**Funding acquisition:** Martin Aberhan.

**Investigation:** Veronica Piazza, Martin Aberhan.

**Methodology:** Veronica Piazza.

**Project administration:** Martin Aberhan.

**Supervision:** Martin Aberhan.

**Visualization:** Veronica Piazza.

**Writing – original draft:** Veronica Piazza, Martin Aberhan.

**Writing – review & editing:** Veronica Piazza, Clemens V. Ullmann, Martin Aberhan.

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
