## [Decision Letter · Decision Letter 0]

10 Jun 2020

PONE-D-20-11804

Temperature affects faunal dynamics of benthic invertebrate assemblages across the Toarcian Oceanic Anoxic Event in the Iberian Basin (Spain)

PLOS ONE

Dear Dr. Piazza,

Thank you for submitting your manuscript to PLOS ONE. After careful consideration, we feel that it has merit but does not fully meet PLOS ONE’s publication criteria as it currently stands. Therefore, we invite you to submit a revised version of the manuscript that addresses the points raised during the review process.

Both reviewers suggest minor revisions and provide excellent comments to improve the ms. Please address all reviewer comments in your reply.

We look forward to receiving your revised manuscript.

Kind regards,

David P. Gillikin, Ph.D.

Academic Editor

PLOS ONE

Journal Requirements:

3. During our internal checks, the in-house editorial staff noted that you conducted research or obtained samples in another country.

Please check the relevant national regulations and laws applying to foreign researchers and state whether you obtained the required permits and approvals.

Please address this in your ethics statement in both the manuscript and submission information.

4. We note that Figure 1 in your submission contains map images which may be copyrighted.

We require you to either (a) present written permission from the copyright holder to publish this figure specifically under the CC BY 4.0 license, or (b) remove the figure from your submission:

b. If you are unable to obtain permission from the original copyright holder to publish this figure under the CC BY 4.0 license or if the copyright holder’s requirements are incompatible with the CC BY 4.0 license, please either i) remove the figure or ii) supply a replacement figure that complies with the CC BY 4.0 license. Please check copyright information on all replacement figures and update the figure caption with source information. If applicable, please specify in the figure caption text when a figure is similar but not identical to the original image and is therefore for illustrative purposes only.

5. In your manuscript, please provide additional information regarding the specimens used in your study.

Ensure that you have reported specimen numbers and complete repository information, including museum name and geographic location.

For more information on PLOS ONE's requirements for paleontology and archaeology research, see https://journals.plos.org/plosone/s/submission-guidelines#loc-paleontology-and-archaeology-research.

Reviewers' comments:

Reviewer's Responses to Questions

**Comments to the Author**

1. Is the manuscript technically sound, and do the data support the conclusions?

Reviewer #1: Yes

Reviewer #2: Yes

2. Has the statistical analysis been performed appropriately and rigorously? 

Reviewer #1: Yes

Reviewer #2: Yes

3. Have the authors made all data underlying the findings in their manuscript fully available?

Reviewer #1: Yes

Reviewer #2: Yes

4. Is the manuscript presented in an intelligible fashion and written in standard English?

Reviewer #1: Yes

Reviewer #2: Yes

5. Review Comments to the Author

Reviewer #1: The work of Piazza et al. 2020 is a thorough and comprehensive look at a locality which represents a better case scenario for marine life trying to survive through the Toarcian extinction event. Rather than the organic-rich shales typically deposited during the severe anoxia of the period, the locality described by the authors exhibited subtler patterns of environmental change, yet still overall displayed a dramatic turnover and transition of the benthic macrofauna known from the site.

The work represents a useful and rigorous case study for researchers expanding our global understanding of biotic change across the Toarcian, an interval of warming and anoxia which caused global disruptions to the carbonate factory paralleling the present crisis our oceans face. I don't have many comments, but present a few opportunities to expand and search for additional lessons from this obviously fruitful fossil locality. No new data would be needed, but these represent a few additional analyses which could bring to light other lessons from your impressive dataset!

68: Here you mention the application of this study to what could be termed conservation paleobiology (using dynamics during past change to better understand our present time), but there is not much follow up. One way I could think of to tie in your results to the present crisis would be to provide a little more detail in the discussion or conclusion on the types of modern biomes the fauna of your site best represent, and how the stressors of the Toarcian are relevant. It could potentially go around line 515 where you start talking about modern community studies. Regarding temperature, it doesn't pass muster for a formal paleotemperature reconstruction but you could use a seawater d18O value of -1 per mil (assuming ice-free world) and as a best-guess taking off point of how the temperature conditions at your site relate to places in the modern day, before and during the extinction event.

260: Many benthic ecologists would hesitate to group true epifaunal taxa (recliners, cementers, etc) with semi-infaunal taxa. Semi-infaunal bivalves often have very distinct needs in terms of substrate grain size, food type, environmental energy level from reclining bivalves, at least calling on my experience with glycymerid bivalves, which can highly stenotopic in their substrate preference. Was this grouping made to improve statistical power? Could another analysis with them separated be run, and reported if it said anything different?

I'd be very interested in how dynamics in occurrence semi-infaunal bivalves relate to pedically attached brachiopods specifically, and also how reclining bivalves relate to the unattached brachiopods.

444: It is perhaps unsurprising that you find that temperature/d18O is the main determinant of faunal change while d13C's influence is messy/absent. I went back to the supplement of the Ullman et al 2020 paper and saw in Supplemental figure S9 that there is an offset between the brachiopod and bivalve results. While combining them is of course the best way to get a large sample size to reconstruct a detailed record of the excursion, perhaps the fact that the excursion itself is poorly correlated to faunal change is not a big shock, as it is a global signal that might not have a huge amount of relevance to the more granular local faunal composition in the way that temperature does.

Could you "un-combine" the record and compare correlation of brachiopod d13C to brachiopod occurrence and vice versa for bivalves? As they can have dietary differences and I'd wonder if there were any dynamics regarding tiering over the interval. You could even see if the two records have an interaction, or subset the brachiopod d13C record by ecology, considering it looks like you have a very large sample size for them. Could be an additional NMDS to run which might help elucidate any ecological trade offs or succession between the groups across the boundary.

500: The energy may have remained relatively low throughout the site, but were there any changes in faunal composition associated with the packstones/rudstones? Indicating possible ecological disruptions from the intermittent storms which required a mini recovery interval? I understand if the resolution you're working with is not enough to make this characterization but might be relevant to those interested in how subtle substrate changes influence the community composition through time. Just bringing up because I had gone back to Ullmann's recent Scientific Reports "Warm Afterglow" paper which described "rhythmic alternations of marlstones and partly argillaceous limestones. The latter primarily comprise mudstones, wackestones, and floatstones." Any faunal associations with these alternations?

I see a couple lines below you say diversity trends are "not consistent" with sea level changes, but could you elaborate on how you made this determination?

Regards and good work,

Dan Killam

Postdoctoral Researcher, Biosphere 2

https://dantheclamman.blog

Reviewer #2: Review of manuscript PONE D-20-11804

Entitled “Temperature affects faunal dynamics of benthic invertebrate assemblages across the Toarcian Oceanic Anoxic Event in the Iberian Basin (Spain) by Piazza et al.

The manuscript for the most part is well written, and provides a lot of detail about one of the many oxygen minimum zones in the Toarcian. The isotope trends are clear markers of this particular event in the Iberian Basin.

I have a few and minor comments that the authors should address before acceptance:

Line 146 d13C -> carbon

213 i.e. -> such as

310 numbers -> …in terms of species and number of specimens.

314 On the contrary -> In contrast; whereas not while, sentence is not clear

318 , while -> whereas species abundance distributions are more even in the post TOAE

326 while ? if at the SAME time, then ok , if not then use ‘whereas’

338 IBID (326)

389 delete ‘down’

427-430 long sentence -please re-write

445 …d18O ‘values’; l. 448 …d13C ‘values’ in isotope terminology the d13C is treated as an adjective and thus needs a ‘noun’

IBID 486, etc

457 …d18O and d13C ‘values’ of what??

444, 474 non—significant -> not significant

524-527 sentence unclear, 527-528 ibid

567 i.e. -> …with cooler temperatures indicating the end..”

L 442 if all d18O and d13C values are based on brachiopods, of short mention in Table 3 and text would be nice.

6. PLOS authors have the option to publish the peer review history of their article (what does this mean?). If published, this will include your full peer review and any attached files.

Reviewer #1: Yes: Daniel Killam

Reviewer #2: Yes: Uwe Brand

---

## [Author Response · Author response to Decision Letter 0]

6 Oct 2020

POINT-BY-POINT REBUTTAL LETTER

Journal Requirements

We have checked the templates and made the few adjustments to meet the journal requirements. Moreover, an additional private email address of the corresponding author V.P. on the title page has been added to ensure continuity in communication in cases of future inquiries about the manuscript. The reason is that the given institutional address will be deactivated in the near future.

More details on the authority that approved of our fieldwork and issued the permit have been given at the end of the “Material and methods” section.

3. During our internal checks, the in-house editorial staff noted that you conducted research or obtained samples in another country. Please check the relevant national regulations and laws applying to foreign researchers and state whether you obtained the required permits and approvals. Please address this in your ethics statement in both the manuscript and submission information.

The original sentence provided in the first version of the manuscript has been modified with the inclusion of the statement provided under Requirement #5. 

4. We note that Figure 1 in your submission contains map images which may be copyrighted. All PLOS content is published under the Creative Commons Attribution License (CC BY 4.0), which means that the manuscript, images, and Supporting Information files will be freely available online, and any third party is permitted to access, download, copy, distribute, and use these materials in any way, even commercially, with proper attribution. For these reasons, we cannot publish previously copyrighted maps or satellite images created using proprietary data, such as Google software (Google Maps, Street View, and Earth). For more information, see our copyright guidelines: http://journals.plos.org/plosone/s/licenses-and-copyright. We require you to either (a) present written permission from the copyright holder to publish this figure specifically under the CC BY 4.0 license, or (b) remove the figure from your submission:

a. You may seek permission from the original copyright holder of Figure 1 to publish the content specifically under the CC BY 4.0 license. We recommend that you contact the original copyright holder with the Content Permission Form (http://journals.plos.org/plosone/s/file?id=7c09/content-permission-form.pdf) and the following text:

Please upload the completed Content Permission Form or other proof of granted permissions as an "Other" file with your submission. In the figure caption of the copyrighted figure, please include the following text: “Reprinted from [ref] under a CC BY license, with permission from [name of publisher], original copyright [original copyright year].”

b. If you are unable to obtain permission from the original copyright holder to publish this figure under the CC BY 4.0 license or if the copyright holder’s requirements are incompatible with the CC BY 4.0 license, please either i) remove the figure or ii) supply a replacement figure that complies with the CC BY 4.0 license. Please check copyright information on all replacement figures and update the figure caption with source information. If applicable, please specify in the figure caption text when a figure is similar but not identical to the original image and is therefore for illustrative purposes only. The following resources for replacing copyrighted map figures may be helpful:

The required permission was requested from the original copyright holder through the RightsLink service. The permission is submitted together with the revised manuscript. The caption of the copyrighted figure was adjusted with the addition of the relevant information.

5. In your manuscript, please provide additional information regarding the specimens used in your study. Ensure that you have reported specimen numbers and complete repository information, including museum name and geographic location.

For more information on PLOS ONE's requirements for paleontology and archaeology research, see https://journals.plos.org/plosone/s/submission-guidelines#loc-paleontology-and-archaeology-research.

Specimen numbers and complete repository information are given at the end of the “Material and methods” section of the manuscript. Details about fieldwork permits are also provided at the end of the same section (see answers to Requirements #2 and #3), with the addition of the statement provided above.

We understand this, and will provide the DOI number as soon as possible upon acceptance of the manuscript.

Comments to the Author

Reviewer #1

The work of Piazza et al. 2020 is a thorough and comprehensive look at a locality which represents a better case scenario for marine life trying to survive through the Toarcian extinction event. Rather than the organic-rich shales typically deposited during the severe anoxia of the period, the locality described by the authors exhibited subtler patterns of environmental change, yet still overall displayed a dramatic turnover and transition of the benthic macrofauna known from the site.

The work represents a useful and rigorous case study for researchers expanding our global understanding of biotic change across the Toarcian, an interval of warming and anoxia which caused global disruptions to the carbonate factory paralleling the present crisis our oceans face. I don't have many comments, but present a few opportunities to expand and search for additional lessons from this obviously fruitful fossil locality. No new data would be needed, but these represent a few additional analyses which could bring to light other lessons from your impressive dataset!

We thank the reviewer for his kind comments and useful insights. His suggestions touch indeed interesting aspects!

68: Here you mention the application of this study to what could be termed conservation paleobiology (using dynamics during past change to better understand our present time), but there is not much follow up. One way I could think of to tie in your results to the present crisis would be to provide a little more detail in the discussion or conclusion on the types of modern biomes the fauna of your site best represent, and how the stressors of the Toarcian are relevant. It could potentially go around line 515 where you start talking about modern community studies. Regarding temperature, it doesn't pass muster for a formal paleotemperature reconstruction but you could use a seawater d18O value of -1 per mil (assuming ice-free world) and as a best-guess taking off point of how the temperature conditions at your site relate to places in the modern day, before and during the extinction event.

We thank the reviewer for this insight. We have added a new paragraph at the end of the discussion comparing early Toarcian and modern faunas, and how the studied event is relevant to modern climate studies and future warming predictions. 

Regarding temperature, we understand that the reviewer’s comment concerns absolute temperature values. Those values are given in our related paper (Ullmann et al. 2020), which includes a discussion of the underlying assumptions. In Figure 3 we kept the more conservative presentation of relative changes in temperature. However, to accommodate the reviewer’s suggestion, we provide absolute temperature estimates in the newly added paragraph mentioned above with reference to the related paper Ullmann et al. (2020). This decision is based on our previous peer-review experience with using absolute temperatures rather than relative temperature change in Ullmann et al. (2020) and Piazza et al. (2020), where reviewers took an adverse stance against absolute temperature estimates, stressing the uncertainties of making inferences on absolute temperatures. There is ongoing debate which of the growing number of published oxygen isotope thermometer equations is the most suitable. Also, absolute temperature reconstructions require assumptions of seawater composition, which are possible, but somewhat uncertain. Given the above limitations, absolute temperature estimates likely have an uncertainty of at least 4°C. Most of this uncertainty is cancelled out when computing temperature change with respect to a reference datum, e.g., the pre-TOAE average. The sensitivity of oxygen isotope thermometers (permil change per °C) is similar and changes only slightly or not at all as a function of temperature in the range of 10 to 30 °C so that the choice of thermometer and the exact position on the temperature scale loose importance. Therefore, the use of relative temperatures allows to assess changing ambient conditions at one site with high confidence and does prevent the (understandable) temptation of comparing absolute temperatures to possible modern analogues.

260: Many benthic ecologists would hesitate to group true epifaunal taxa (recliners, cementers, etc) with semi-infaunal taxa. Semi-infaunal bivalves often have very distinct needs in terms of substrate grain size, food type, environmental energy level from reclining bivalves, at least calling on my experience with glycymerid bivalves, which can highly stenotopic in their substrate preference. Was this grouping made to improve statistical power? Could another analysis with them separated be run, and reported if it said anything different? 

I'd be very interested in how dynamics in occurrence semi-infaunal bivalves relate to pedically attached brachiopods specifically, and also how reclining bivalves relate to the unattached brachiopods.

We understand and agree with this observation, and we thank the reviewer for pointing this out. Indeed, the main reason for lumping semi-infaunal and fully epifaunal taxa, both being exposed to potential predators at the sea floor, was to increase statistical power. Only ca. 10% of the specimens belong to the semi-infauna (see MOL 7 in Fig. 7) – too few to be considered on its own. To see the influence of the semi-infauna we followed the reviewer’s suggestion and analysed the two life-habit categories separately. Overall, the results obtained for epifauna + semi-infauna and for true epifauna are congruent (i.e. no significant changes). To address the concerns of the reviewer, we now present the results for the true epifauna instead of the lumped epifauna + semi-infauna. The text, Table 2 and Fig. 9 have been modified accordingly. 

As for semi-infaunal bivalves vs. brachiopods, we feel there is not much to add in this respect. The important information was already presented in S2 Fig. To make this more evident we now moved previous S2 Fig to the main manuscript where it became the new Fig. 7. It shows the relative abundance change of all MOLs including semi-infaunal bivalves (MOL 7) and pedically attached brachiopods (MOL 2). The only other realised life habit in brachiopods is represented by the shallow infaunal attached Lingularia sp. (MOL 14) which is very rare. No unattached brachiopods are present in the studied fauna as the reviewer seems to imply.

444: It is perhaps unsurprising that you find that temperature/ δ18O is the main determinant of faunal change while δ13C's influence is messy/absent. I went back to the supplement of the Ullman et al 2020 paper and saw in Supplemental figure S9 that there is an offset between the brachiopod and bivalve results. While combining them is of course the best way to get a large sample size to reconstruct a detailed record of the excursion, perhaps the fact that the excursion itself is poorly correlated to faunal change is not a big shock, as it is a global signal that might not have a huge amount of relevance to the more granular local faunal composition in the way that temperature does.

Could you "un-combine" the record and compare correlation of brachiopod δ13C to brachiopod occurrence and vice versa for bivalves? As they can have dietary differences and I'd wonder if there were any dynamics regarding tiering over the interval. You could even see if the two records have an interaction, or subset the brachiopod δ13C record by ecology, considering it looks like you have a very large sample size for them. Could be an additional NMDS to run which might help elucidate any ecological trade offs or succession between the groups across the boundary.

The reviewer has made a series of quite interesting points. "Uncombining" the carbon isotope records of bivalves and brachiopods can easily be done as all the raw data are available and are listed in the supplements of Ullmann et al. (2020), but we don't think that it would yield significant further insights. Bivalves are invariably enriched in 13C versus coeval brachiopods and most enrichment factors agree with the average within uncertainty. Only in the range of 5 to 7 m height in the section the values are outside 95 % confidence, but this seems to be a part of the section where nothing really noteworthy changes regarding assemblages. Even though overall the dataset is large, taking the dataset apart might lead to over-interpreting some of the results because we are dealing with signals much smaller than 0.5 permil, if they are present at all. We feel that the safest approach - given the heterogeneity of macrofossil datasets - is stating that brachiopod and bivalve carbon isotope records run parallel with an approximately 0.6 permil offset towards higher δ13C in bivalves.

It may be useful to revisit the observation of bivalve 13C enrichment (but missing 18O offset) and to try to find a satisfactory explanation rather than keeping it at the pragmatic solution we employed then, but so far it seems to us that we (the community in general that is) don't really know enough about exactly how these different organisms form shell material and what this may mean for isotopic signatures.

Regarding inter-species differences, there are no clear differences between any of the rhynchonellid genera (they are certainly not larger than between individuals of the same species and this point has been explored in detail in the supplements of Ullmann et al., 2020). We would hesitate making environmental inferences from comparison to terebratulid data even if we had sufficient data to do so because terebratulid vital effects make this too prone to biased outcomes. Bivalve data are Gryphaea-only, so it would be impossible to look into finer detail there.

500: The energy may have remained relatively low throughout the site, but were there any changes in faunal composition associated with the packstones/rudstones? Indicating possible ecological disruptions from the intermittent storms which required a mini recovery interval? I understand if the resolution you're working with is not enough to make this characterization but might be relevant to those interested in how subtle substrate changes influence the community composition through time. Just bringing up because I had gone back to Ullmann's recent Scientific Reports "Warm Afterglow" paper which described "rhythmic alternations of marlstones and partly argillaceous limestones. The latter primarily comprise mudstones, wackestones, and floatstones." Any faunal associations with these alternations? I see a couple lines below you say diversity trends are "not consistent" with sea level changes, but could you elaborate on how you made this determination?

This is a valid point. We are aware of the influence that substrate changes can have on faunal composition. Yet, our samples are derived from fairly homogenous lithologies (wackestones and floatstones with just one sample frpm a rudstone. We now made this clear in the manuscript text. We also performed again the cluster and NMDS analyses on both taxonomical and functional composition of the studied faunas, including lithology as a grouping factor to investigate whether faunal associations related to specific lithologies could be observed. It is quite apparent that lithological differences did not have any larger effect on faunal composition. These new results are briefly mentioned in the text and the clusters in Supplementary Figure 1 were adjusted to include lithological differences.

Regarding diversity trends in relation to sea level change, we have made adjustments to make clearer the basis of our statement.

Reviewer #2

The manuscript for the most part is well written, and provides a lot of detail about one of the many oxygen minimum zones in the Toarcian. The isotope trends are clear markers of this particular event in the Iberian Basin. I have a few and minor comments that the authors should address before acceptance:

We thank the reviewer for his positive assessment. As illustrated in detail below, we have addressed most of the comments, and justified the few ones we decided not to incorporate in the revised manuscript.

Line 146 δ13C -> carbon 

Done.

213 i.e. -> such as

The Reviewer suggested substitution of “i.e.” with “such as”. We feel that it would imply the explanation for MOL given here as an example, while we provide a definition. We use then “–“ instead.

310 numbers -> …in terms of species and number of specimens.

Done.

314 On the contrary -> In contrast; whereas not while, sentence is not clear. 

Done.

318 , while -> whereas species abundance distributions are more even in the post TOAE 

Done.

326 while ? if at the SAME time, then ok , if not then use ‘whereas’

As in that sentence we are still discussing the same time interval (the pre-TOAE), we keep “while” instead of “whereas”.

338 IBID (326)

In this instance we changed “while” into “whereas”.

389 delete ‘down’

Done.

427-430 long sentence -please re-write

We have rearranged the sentence.

445 … δ18O ‘values’; l. 448 … δ13C ‘values’ in isotope terminology the d13C is treated as an adjective and thus needs a ‘noun’

We have adjusted it here, and also when needed elsewhere in the text.

IBID 486, etc

We did not see any mention of isotopes at line 486 of the originally submitted manuscript. This comment probably referred to line 468 and we adjusted the text here and elsewhere in the manuscript.

457 … δ18O and d δ13C ‘values’ of what?? 

Adjusted.

444, 474 non—significant -> not significant 

Done.

524-527 sentence unclear, 527-528 ibid

Sentences rearranged to make the content clear.

567 i.e. -> …with cooler temperatures indicating the end..”

Done.

L 442 if all δ18O and δ13C values are based on brachiopods, of short mention in Table 3 and text would be nice.

As stated in the methods, isotope data are derived from both oysters and brachiopods. We added this information again in the caption of Table 3, and feel therefore that no further change in the text is required.

---

## [Editor Report · Decision Letter 1]

2 Nov 2020

Ocean warming affected faunal dynamics of benthic invertebrate assemblages across the Toarcian Oceanic Anoxic Event in the Iberian Basin (Spain)

PONE-D-20-11804R1

Dear Dr. Piazza,

We’re pleased to inform you that your manuscript has been judged scientifically suitable for publication and will be formally accepted for publication once it meets all outstanding technical requirements.

Kind regards,

David P. Gillikin, Ph.D.

Academic Editor

PLOS ONE

---

## [Editor Report · Acceptance letter]

17 Nov 2020

PONE-D-20-11804R1 

Ocean warming affected faunal dynamics of benthic invertebrate assemblages across the Toarcian Oceanic Anoxic Event in the Iberian Basin (Spain) 

Dear Dr. Piazza:

I'm pleased to inform you that your manuscript has been deemed suitable for publication in PLOS ONE. Congratulations! Your manuscript is now with our production department. 

Kind regards, 

on behalf of

Dr David P. Gillikin 

Academic Editor

PLOS ONE